# SteP: Stacked LLM Policies for Web Actions

**Paloma Sodhi**[1][*], **S.R.K. Branavan**[1], **Yoav Artzi**[1,2], **Ryan McDonald**[1]
[1]ASAPP Research, NY, USA
[2]Cornell University, NY, USA
{psodhi, branavan, rmcdonald}@asapp.com, yoav@cs.cornell.edu

## Abstract

Performing tasks on the web presents fundamental challenges to large language models (LLMs), including combinatorially large open-world tasks and variations across web interfaces. Simply specifying a large prompt to handle all possible behaviors and states is extremely complex, and results in behavior leaks between unrelated behaviors. Decomposition to distinct policies can address this challenge but requires carefully handing off control between policies. We propose Stacked LLM Policies for Web Actions (`SteP`), an approach to dynamically compose policies to solve a diverse set of web tasks. `SteP` defines a Markov Decision Process where the state is a *stack of policies* representing the control state, i.e., the chain of policy calls. Unlike traditional methods that are restricted to static hierarchies, `SteP` enables *dynamic* control that adapts to the complexity of the task. We evaluate `SteP` against multiple baselines and web environments including WebArena, MiniWoB++, and a CRM. On WebArena, `SteP` improves (14.9% to 33.5%) over SOTA that uses GPT-4 policies, while on MiniWob++, SteP is competitive with prior works while using significantly less data. Our code and data are available at https://asappresearch.github.io/webagents-step.

## 1 Introduction

While large language model (LLM) agents have shown impressive decision-making capabilities (Yao et al., 2022b; Huang et al., 2022b), the web remains a challenging domain achieving much lower success rates compared to other benchmarks (Akter et al., 2023; Zhou et al., 2023). The web contains a combinatorially large open-world space of tasks such as booking flights, purchasing items or making appointments. Web interfaces also differ substantially from one website to another, for instance, the task of purchasing an item on Amazon looks different from purchasing it on eBay.

There are fundamental challenges to designing a singular LLM policy to solve all possible web tasks. First, the policy requires instructions and examples to cover all variations in tasks and websites. Second, solving longer horizon tasks requires keeping around a long history of previous actions and observations in context. Longer contexts make it harder to pay attention to salient information leading to more errors and costs (Liu et al., 2023).

Instead, a natural solution is to decompose the problem into distinct policies (Khot et al., 2023). Each policy provides dedicated instructions and examples for a particular subproblem, such as searching a list or finding a page. However, this typically requires manually specifying a decomposition hierarchy that hands off control between policies (Prasad et al., 2023; Zhou et al., 2021; Song et al., 2023). This restricts control to a static hierarchy that fails to adapt to varying task complexity.

*Our key insight is to enable dynamic control, where any policy can choose to invoke any other policy.* Such expressiveness is crucial for solving web tasks that require policies to operate at multiple levels of abstraction. Consider Fig. 1 where the agent must find all commits

---

[*]Corresponding author: Paloma Sodhi <psodhi@asapp.com>

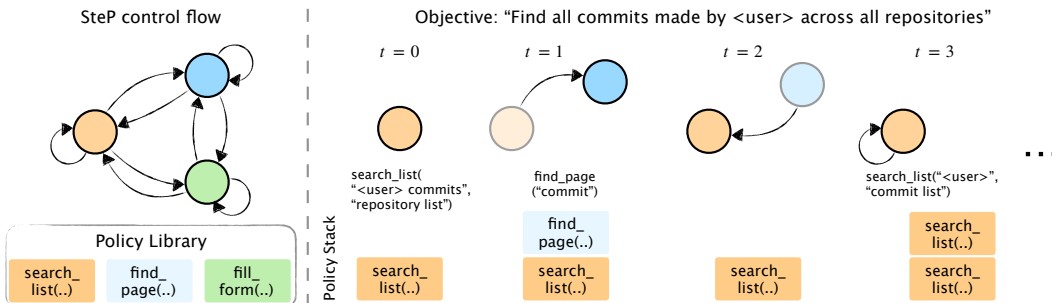

Figure 1: `SteP` composes policies to solve complex task, where policies can invoke each other. `SteP` uses a policy stack to keep track of the dynamic control state. Given an objective "Find all commits made by a <user> across all repositories", `SteP` intializes with a `search_list()` to search over all repositories, which in turn invokes another `search_list()` to search overall commits in a repository.

made by a user across all repositories. A `search_list()` policy must first iterate over all repositories. For every repository, it must recursively call another `search_list()` that iterates over all commits in that repository. This can only be solved by an architecture where policies can call each other, including themselves.

We propose **S**tacked LLM **P**olicies for Actions on the Web (`SteP`), a method to perform a diverse set of web tasks by dynamically composing policies. `SteP` defines a Markov Decision Process (MDP) where the state is a stack of policies. The stack stores the dynamic control state capturing the chain of policy calls that evolves over time. At every time step, the policy on the top of the stack either acts directly on the web page, invokes a new policy that gets pushed onto the stack, or terminates and pops out of the stack. For instance, in Fig. 1 task, the stack initializes with a `search_list()` policy, which can both act on the web page or instantiate another `search_list()` policy with a different set of arguments.

Our key findings are that dynamically composing policies (`SteP`) significantly outperforms both prior works ($0.15 \rightarrow 0.33$) and single policy baselines ($0.23 \rightarrow 0.33$). `SteP` achieves this while using 2.3x lesser tokens per trajectory, resulting in lower overall costs. We also show several ablations on the effect of varying contexts, in-context examples, and CoT reasoning.

Our key contributions are:

1. A novel framework `SteP` that defines an MDP over a stack of policies enabling dynamic composition of policies to solve complex web tasks.
2. Experimental validation on a range of web benchmarks: WebArena, MiniWoB++, and an airline CRM simulator. On WebArena, `SteP` improves ($0.15 \rightarrow 0.33$) over prior works that use few-shot LLM (GPT-4) policies, while on MiniWob++, `SteP` is competitive with prior works while using significantly less data.
3. Implementation of `SteP` as a meta-policy that wraps around any existing policy class.

## 2 Related Work

**Language models for web tasks.** Early work mapping natural language instructions into actions (Branavan et al., 2009; Artzi & Zettlemoyer, 2013) has rapidly evolved into the field of LLM agents (Wang et al., 2023b). Broadly, methods include training RL agents to navigate web interfaces (Humphreys et al., 2022; Liu et al., 2018; Shi et al., 2017), in-context learning with large language models (Zhou et al., 2023; Zheng et al., 2024; Kim et al., 2023), or finetuning language models on web tasks (Deng et al., 2023; Furuta et al., 2024; Yao et al., 2022a). With in-context learning, using a single LLM policy that contains all instructions and examples results in long contexts that can be error-prone. Recent approaches (Zheng et al., 2024; Kagaya et al., 2024) counter this by retrieving trajectories from a database. However, covering the combinatorial space of tasks requires a large dataset from many tasks. Instead, our work leverages a library of policies, each with dedicated instructions and examples, and composes policies to cover such tasks.

**Language models for decision making.** Instruction following LLMs have shown impressive decision making capabilities (Huang et al., 2022a; Brown et al., 2020) and tool use (Schick et al., 2023; Yang et al., 2024) by chaining together reason and actions (Yao et al., 2022b). However, for long-horizon tasks, a single policy with a long chain of reason and action can be error-prone. Broadly works deal with such issues by hierarchical planning (Prasad et al., 2023; Khot et al., 2023; Zhou et al., 2021), defining state machines (Ma et al., 2023), generating code (Wang et al., 2023a; Liang et al., 2022), or by self-correction (Shinn et al., 2023). Hierarchical decision-making has a rich history in AI (Sutton, 1998), where a high-level policy chooses either from a library of skills (Tessler et al., 2017), or predicts subgoals (Nachum et al., 2018) or predicts rewards (Vezhnevets et al., 2017) for a low-level policy. While these methods focus on efficiently learning policies, they restrict the decomposition to a predefined hierarchy of 2 or 3 levels. Instead, our framework allows any policy to dynamically call any other policy in the library, with control state being tracked using a policy stack. Compared to decomposition approaches like DecomP (Khot et al., 2023) which generates a static program with subroutines that cannot invoke each other, SteP decomposes dynamically based on observations and allows policies to invoke each other. Compared to hierarchical planning approaches like ADaPT (Prasad et al., 2023), which predicts a plan, SteP instead predicts policies that can react to failures more quickly.

**Automata and Transition-based Parsing.** The fact that the main control structure of our formalism is a stack relates our algorithm to Pushdown Automata (Hopcroft & Ullman, 1969). However, in Pushdown Automata, the input tape is static and processed sequentially. In contrast, for web actions the input tape is dynamic due to new observations that arise. These observations dynamically alter the context (or input string) that the policies work with. While different in scope, our work does draw inspiration from transition-based dependency parsing systems, specifically stack-based algorithms (Nivre, 2008). In particular, our algorithm consists of states whose main data structure is a stack and states transition to new states via a finite set of well defined actions (see Sec. 4.1).

## 3  Problem Formulation

Given a natural language instruction, such as "Book me a flight from NYC to BOS", our objective is to learn a policy $\pi$ that execute this task on a web environment. This can be formulated as a Partially Observable Markov Decision Process (POMDP), denoted as $\langle \mathcal{S}, \mathcal{A}, \mathcal{O}, \mathcal{T}, r \rangle$. At each time step $t$, the policy (or agent) performs an action $a_t$ given a partial observation $o_t$, resulting in a new state $s_{t+1}$ and observation $o_{t+1}$, where:

- **State,** $s \in \mathcal{S}$ is the current state of the web environment, including the current webpage contents and results from previous interactions, e.g., a new repository created in a GitLab environment.
- **Action,** $a \in \mathcal{A}(s)$ denotes the possible actions that can be performed in the current state, such as clicking, scrolling, typing on specific web elements. These are represented as `click [id]`, `type [id] [value]`, where id refers to a specific element on the webpage. The action space is typically large due to the many elements on a web page.
- **Observation,** $o \in \mathcal{O}$ is the current observable aspect of the state, i.e., the current webpage Document Object Model (DOM) serialized as text.
- **Transition function,** $\mathcal{T}(s'|s, a)$ is a deterministic function modeling the change in the webpage resulting from an action determined by the underlying website.
- **Reward,** $r(s, a)$ is awarded for reaching a set of subgoals, e.g. canceling a flight has subgoals like finding the booking and then canceling it.

As the state is partially observable, the policy maps a history of observations and actions $h_t = \{o_t, a_{t-1}, o_{t-1}, ..\}$ to the current action $a_t$, i.e. $\pi : h_t \rightarrow a_t$. As discussed before, learning a single LLM web policy $\pi$ is challenging. We next look at composing multiple policies.

## 4  Approach

We present a framework, **S**tacked LLM **P**olicies for Web Actions (`SteP`), that performs a range of web tasks by dynamically composing policies. As previously discussed, designing a single policy that solves all tasks is challenging. Instead, we utilize a library of policies $\Pi$ which we compose to solve a task. Fig. 2 shows an illustration of `SteP` solving a web task

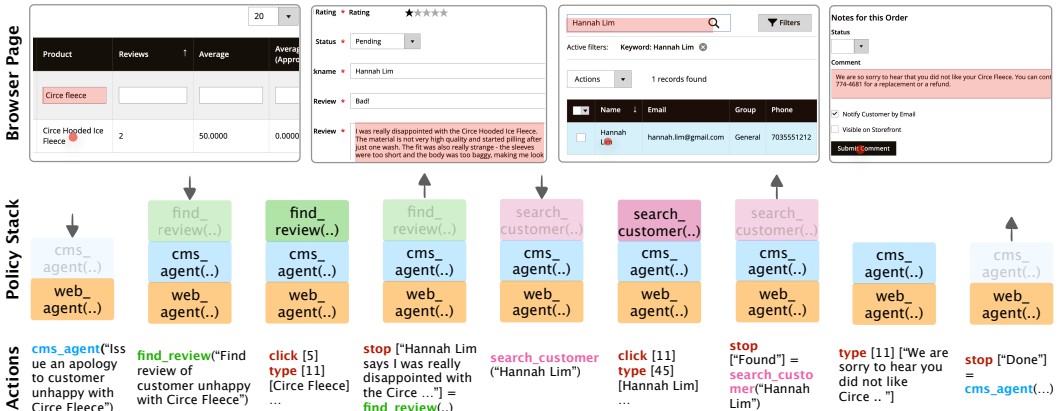

Figure 2: Example of SteP solving a web task on a Customer Management System (CMS) website. SteP dynamically composes policies from a library, using a policy track to keep track of active policies. At every timestep, SteP either acts on the webpage, or modifies the stack to add/remove policies.

by dynamically stacking policies from the library. Sec. 4.1, 4.2 discusses the stacked policy model and SteP algorithm respectively.

## 4.1 Stacked Policy Model

At any given time, we represent the control state as a stack $\Sigma$ which denotes the chain of policy calls $\Sigma = \langle \pi_0 | \pi_1 | \ldots | \pi_i \rangle$. We extend the MDP in Sec. 3 to include a stack of policies:

**State.** The MDP state is augmented with a stack $\Sigma = \langle \pi_0 | \pi_1 | \ldots | \pi_i \rangle$ of invoked policies. The top of the stack $\pi_i$ is the current active policy. Each policy in the stack maintains its own history of observation, reason, action. The stack is initialized with a base policy $\Sigma = [\pi_0]$.

**Action.** We augment the original MDP actions with two new actions – invoke a new policy $\pi \in \Pi$ or terminate the current policy $\pi_i$ with a return value $v_i$.

**Transition.** Suppose at timestep $t$, the current stack is $\Sigma_t = \langle \pi_0 | \pi_1 | \ldots | \pi_i[h] \rangle$. The policy on top of the stack, $\pi_i$, can take one of three actions leading to different state transitions:

1. *Issue an action:* It can issue an action $a_t$ along with reason $r_t$. This is sent to the environment, which updates its state $s_{t+1}$ and responds with an observation $o_{t+1}$. The action, reason, and observation is appended to the history maintained by $\pi_i$. The set of policies in the stack remains unchanged, only the history for the current policy updates.

$$\langle \pi_0 | \pi_1 | \ldots | \pi_i[h] \rangle \rightarrow \langle \pi_0 | \pi_1 | \ldots | \pi_i[h \leftarrow h \cup (a_t, r_t, o_{t+1})] \rangle \qquad (1)$$

2. *Invoke another policy:* It can choose to invoke a new policy $\pi_{i+1}$. The new policy is initialized with an empty history and is pushed onto the stack.

$$\langle \pi_0 | \pi_1 | \ldots | \pi_i \rangle \rightarrow \langle \pi_0 | \pi_1 | \ldots | \pi_i | \pi_{i+1} \rangle \qquad (2)$$

   No action is sent to the environment.

3. *Terminate and hand back control:* It can choose to terminate with a return value $v_i$ which in our case is an optional response returned by the policy once it finishes executing. The policy $\pi_i$ is popped off the stack, and the response is added to the history of $\pi_{i-1}$.

$$\langle \pi_0 | \pi_1 | \ldots | \pi_{i-1}[h] | \pi_i \rangle \rightarrow \langle \pi_0 | \pi_1 | \ldots | \pi_{i-1}[h \leftarrow h \cup v_i] \rangle \qquad (3)$$

   No action is sent to the environment.

**Reward.** The reward functions are the same as the original MDP.

While this is a novel decision making model, it has close ties to transition-based dependency parsing systems in NLP and hierarchical decision making in RL. See Sec. 2.

**Algorithm 1** `SteP` : Dynamically compose policies to solve a web task

```python
class SteP(Policy):
    ...
    def predict_action(self, observation):
        if self.stack.is_empty():
            root_policy = self.init_policy()
            self.stack.push(root_policy)

        while not self.stack.is_empty():
            policy = self.stack.top()
            action, reason = policy.predict_action(observation)
            # Issue an environment action
            if self.is_environment_action(action):
                return action, reason
            # Invoke a new policy
            if self.is_policy_action(action):
                new_policy = self.init_policy(action)
                self.stack.push(new_policy)
                continue
            # Terminate and hand back control
            if self.is_policy_done(action):
                self.stack.pop()
                policy = self.stack.top()
                policy.append_response(action) if policy else None
                continue
        return action, reason #Termination action by root policy

def main()
    policy = SteP()
    observation, done = env.reset(), False
    while not done:
        action, reason = policy.predict_action(observation)
        observation, done = env.step(action)
```

### 4.2 SteP Algorithm

Algorithm 1 presents the pseudo-code for `SteP`. `SteP` is a meta-policy that wraps around a typical `Policy` class. `SteP` maintains a `self.stack` variable to store the set of active policies. We focus on the `predict_action()` that takes as input the observation and returns an action and a reason. It begins by initializing the stack with a root policy. It then takes the policy at the top of the stack and invokes its `predict_action()` function.

The policy can perform one of three actions. First, if the action is an environment action, e.g. `click [id]`, it returns that directly. Second, if the action is a call to another policy, e.g. `search_customer(..)`, it initializes the corresponding policy, pushes it to the top of the stack, and assigns it control. Third, if the action indicates that the current policy is done acting, e.g. `stop [response]`, it pops the current policy out of the stack, sends the response to the next policy on the stack, and assigns it control.

**Key features.** Several characteristics emerge from such an approach, notably:

1. *Dynamic Composition.* Policies are composed dynamically at test time based on observations from the environment. The space of possible control states are defined by each policy's action space, i.e. other policies they can transition to. The stack can having a varying depth that adapts to task difficulty. Compared to prior works (Akter et al., 2023; Zhou et al., 2023) that use a single policy, composition allows for more adaptability.
2. *Scalability.* Adding a new policy to `SteP` is easy. The user constructs the prompt for the policy and adds it to the library with a description. This policy becomes available as part of the action space for any other policy, without requiring any change to the code.
3. *Modularity.* Each policy tracks only the local context of the specific subproblem it is solving. Once it terminates, it hands back control to the previous policy in the stack without reasoning about the global context. This allows reusing policies in different context, e.g. the same `fill_form()` can be used by book flights or make appointments. Compared to prior work (Zheng et al., 2024) that require demonstration for entire tasks, modularity requires demonstrations for subtasks thus being more sample efficient.

## 5 Experiments

### 5.1 Experimental Setup

**Environments.** We evaluate across multiple distinct web environments listed below.

- **WebArena** (Zhou et al., 2023). A recent benchmark with complex web tasks across multiple domains like shopping, software development, content management. WebArena websites are highly realistic with tasks mirroring those that humans routinely perform on the internet. We evaluate across all 804 tasks in the benchmark.
- **MiniWoB++** (Liu et al., 2018; Shi et al., 2017). Compared to WebArena, this is a simplified web environment covering interactions like form filling, search, choosing dates. We evaluate across all 45 tasks that don't rely on vision and average over 50 seeds per task.
- **AirlineCRM**. We develop a new CRM simulator (Appendix C) modeled after customer service workflows on popular airline websites. Compared to MiniWoB++, this contains longer-horizon tasks. We evaluate across 5 tasks averaged over 20 scenarios per task.
- Finally we test on live website environments and show results in Appendix D.

**Policies.** We use a library of 14 policies for WebArena, each covering multiple intents. Constructing a policy is straightforward: we use a templated prompt with general instructions, action space definition, and place holders for policy specific instructions and examples. To design policies, we cluster intents that are functionally equivalent, e.g., searching over orders or listing products. See Appendix A.2.

**Baselines.** We compare against various baselines including prior state-of-the-art on WebArena (Zhou et al., 2023; Akter et al., 2023) which design a single web agent policy following a ReAct (Yao et al., 2022b) style chain-of-thought (CoT) prompting. On MiniWob++, we compare against recent fine-tuning (Furuta et al., 2024; Gur et al., 2022a; Humphreys et al., 2022) and in-context learning (Zheng et al., 2024; Sun et al., 2024; Kim et al., 2023) works.

Additionally, we create baselines to study the following effects: (i) *Single vs Decomposed prompt* (Flat vs SteP). Flat is a single policy that concatenates instructions and examples from all the policies in the library into a single prompt. (ii) *Varying context lengths of prompts* (Flat-4k vs Flat-8k). Since the typical policy prompt is less than 4000 tokens, we create two baselines Flat-4K that caps the prompt size to 4000 tokens and Flat-8K that caps it to 8000. (iii) *Effect of in-context examples* (Zero-shot vs Few-shot). We study effect of adding observation action examples that help associate language instructions to webpage elements.

We study **(i)** on all datasets, **(ii)** on WebArena, where context lengths are longer due to more complex webpages, and **(iii)** on MiniWob++, AirlineCRM where the stripped down webpages have ambiguous elements (e.g. missing aria-labels) and benefit from examples.

**Models.** For models, on WebArena we evaluate with gpt-4-turbo[1](OpenAI, 2023) since the tasks are complex, while for MiniWob++ and AirlineCRM we evaluate with either the instruction fine-tuned text-davinci-003[1] or gpt-3.5-turbo[1](Ouyang et al., 2022).

**Metrics.** We define 3 metrics: Success Rate (suc↑), Task Progress (prog↑), and Number Actions (#act). suc↑ is either 0 or 1 depending on the task being completed successfully. #act is the number of actions taken. On airline CRM, we also compute prog↑ a number between 0 and 1 indicating progress towards completing the task.

### 5.1.1 Overall Results

- On WebArena, SteP outperforms prior works (0.15 → 0.33) with improvements on every environment: Shopping (0.2 → 0.37), CMS (0.10 → 0.24), Reddit (0.11 → 0.59), Gitlab (0.14 → 0.32, Maps (0.15 → 0.30). See Sec. 5.1.2.
- SteP achieves greater accuracy over Flat-8k (0.23 → 0.33) while using 2.3x smaller context lengths per episode. See Sec. 5.1.3, 5.1.4.
- On MiniWob++, SteP Few-shot is competitive to prior works while using significantly less data. See Sec. 5.1.2.

---

[1]https://platform.openai.com/docs/models

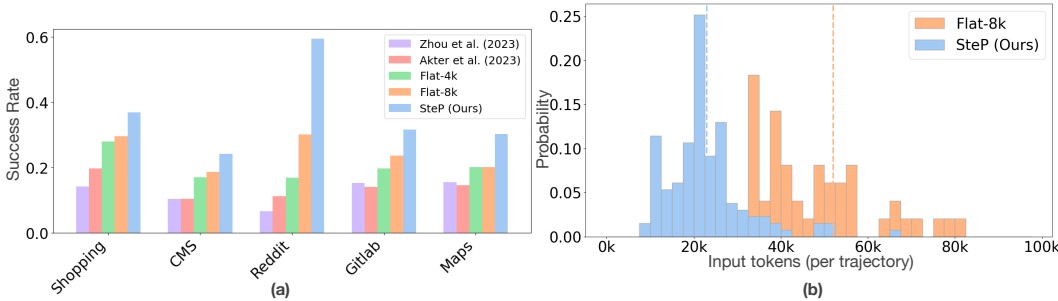

Figure 3: **(a)** Success rates of `SteP` against all baselines on 5 different WebArena websites. **(b)** Distribution of input tokens per trajectory of `SteP` vs `Flat-8k` on WebArena. `SteP` achieves higher success rates while needing less input tokens resulting in lower costs per trajectory.

| | Sampled Task Intents | Zhou et al. (2023) | Akter et al. (2023) | Flat-4k | | Flat-8k | | SteP | |
|---|---|---|---|---|---|---|---|---|---|
| | (samples per website) | suc↑ | suc↑ | suc↑ | #act | suc↑ | #act | suc↑ | #act |
| **Shopping** | List customers complaint about {items} | 0.00 | 0.00 | 0.00 | 4.00 | 0.00 | 4.00 | 1.00 | 15.0 |
| | Config of the {product} I bought {time} | 0.20 | 0.00 | 0.20 | 11.0 | 0.00 | 4.20 | 0.60 | 11.6 |
| | Show most recent {status} order | 0.20 | 0.00 | 0.40 | 4.40 | 0.40 | 4.00 | 0.20 | 2.00 |
| | Summarize main criticisms of product | 0.00 | 0.20 | 0.60 | 2.00 | 0.40 | 2.20 | 0.00 | 10.4 |
| | Show {product} listings by { order} | 0.20 | 0.20 | 0.20 | 2.00 | 0.20 | 2.17 | 0.20 | 12.2 |
| | **Mean all 48 Shopping intents** | **0.14** | **0.20** | **0.28** | 6.66 | **0.30** | 4.83 | **0.37** | 8.99 |
| **CMS** | Update order #{order} with tracking | 0.00 | 0.00 | 0.00 | 1.20 | 0.00 | 4.40 | 0.20 | 15.0 |
| | Tell reasons customers like {product} | 0.00 | 0.00 | 0.00 | 1.20 | 0.20 | 4.80 | 0.00 | 8.80 |
| | Notify {name}: {message} | 0.00 | 0.00 | 0.40 | 1.60 | 0.40 | 3.80 | 0.80 | 11.6 |
| | Find customer by {PhoneNum} | 0.20 | 0.00 | 0.60 | 3.00 | 0.40 | 5.00 | 0.40 | 4.60 |
| | How many reviews received on {time} | 0.20 | 0.60 | 0.60 | 3.20 | 0.60 | 2.80 | 0.20 | 6.80 |
| | **Mean all 41 CMS intents** | **0.11** | **0.10** | **0.17** | 1.75 | **0.18** | 5.42 | **0.24** | 11.6 |
| **Reddit** | Post { notice } in {subreddit} | 0.00 | 0.00 | 0.00 | 13.2 | 0.00 | 20.0 | 0.60 | 7.80 |
| | Like submissions {user} in {subreddit} | 0.00 | 0.17 | 0.50 | 4.67 | 0.33 | 9.83 | 0.50 | 14.5 |
| | Create new {forum} with {description} | 0.00 | 0.20 | 0.60 | 7.00 | 0.40 | 0.76 | 0.80 | 9.00 |
| | Post review {book} with {comment}. | 0.20 | 0.20 | 0.20 | 10.8 | 0.80 | 17.60 | 0.90 | 8.00 |
| | Reply to {post} with {content} | 0.20 | 0.60 | 0.60 | 9.00 | 0.60 | 13.6 | 0.00 | 3.00 |
| | **Mean all 21 Reddit intents** | **0.06** | **0.11** | **0.17** | 9.86 | **0.30** | 12.09 | **0.59** | 9.26 |
| **Gitlab** | Create new {group} with {members} | 0.00 | 0.00 | 0.00 | 9.80 | 0.00 | 10.0 | 0.80 | 14.2 |
| | Commits {user} made to {repo}? | 0.20 | 0.20 | 0.40 | 3.00 | 0.40 | 3.00 | 0.80 | 3.80 |
| | Check latest issue {keyword} if closed | 0.00 | 0.00 | 0.00 | 2.80 | 0.00 | 3.00 | 0.20 | 6.80 |
| | Check out the most recent open issues | 0.00 | 0.00 | 0.00 | 5.00 | 0.50 | 2.50 | 0.50 | 2.00 |
| | Open an issue to {issue} in {repo} | 0.17 | 0.33 | 0.33 | 4.40 | 0.33 | 7.00 | 0.80 | 10.4 |
| | **Mean all 41 Gitlab intents** | **0.15** | **0.14** | **0.20** | 5.48 | **0.23** | 6.65 | **0.32** | 9.34 |
| **Maps** | Nearest {location} from {location2} | 0.00 | 0.00 | 0.00 | 18.0 | 0.00 | 20.0 | 0.00 | 6.00 |
| | Closest {place1}(s) to {place2} | 0.00 | 0.00 | 0.00 | 17.0 | 0.20 | 16.8 | 0.60 | 10.8 |
| | Driving time {city1} to {city2} | 0.00 | 0.00 | 0.75 | 4.75 | 0.75 | 4.50 | 0.75 | 5.00 |
| | From {hotel}, time to reach {place} | 0.20 | 0.40 | 0.40 | 5.00 | 0.40 | 5.00 | 0.40 | 8.20 |
| | Find the {space} around {location} | 0.40 | 0.20 | 0.40 | 10.6 | 0.40 | 10.0 | 0.40 | 6.20 |
| | **Mean all 29 Maps intents** | **0.16** | **0.15** | **0.20** | 6.89 | **0.20** | 7.64 | **0.30** | 11.2 |
| | **Mean all WebArena intents** | **0.12** | **0.15** | **0.20** | 6.44 | **0.23** | 7.25 | **0.33** | 10.0 |

Table 1: **WebArena Success Rates** over 804 tasks categorized by task intents across different websites. Values shown for 5 intents sampled from 5 quantiles, average values shown for all intents for each website. Each intent consists of 3-5 tasks. `SteP` caps each policy prompt to 4k tokens.

- In-context examples help in addition to instructions by associating language instructions with corresponding web elements. Moreover, `SteP Few-shot` uses these examples more effectively by having them in dedicated policies. See Section 5.1.5.
- We provide ablations on CoT reasoning and model scales in Appendix B.

### 5.1.2 Comparison to prior works

On WebArena, Fig. 3(a), Table 1 show that `SteP` outperforms prior works (Zhou et al., 2023; Akter et al., 2023) that use a single GPT-4 policy on all environments. By having a library of only 14 policies (see Appendix A.2), `SteP` cov-

ers at least 50 intents out of a total of 170 intents. Most significant gains come from Shopping, Reddit where the policies cover a larger percentage of intents.

On MiniWob++, Table 2 shows that SteP outperforms all baselines that fine-tune models. It uses only 10 demonstration trajectories (24 observation-action examples) compared to the most recent baseline (Furuta et al., 2024) that trains on 347K trajectories. It is also competitive to recent in-context learning works while using fewer trajectories. For instance, Synapse (Zheng et al.,

| Method | Models | Training trajectories | Success Rate |
|---|---|---|---|
| WGE (Liu et al., 2018) | - | 12K+ | 0.76 |
| CC-Net (SL) (Humphreys et al., 2022) | ResNet | 2.4M | 0.36 |
| CC-Net (SL+RL) (Humphreys et al., 2022) | ResNet | 2.4M | 0.96 |
| WebN-T5 (Gur et al., 2022b) | T5-XL | 12K | 0.56 |
| WebGUM (HTML) (Furuta et al., 2024) | Flan-T5-XL | 347K | 0.90 |
| RCI (Kim et al., 2023) | gpt-4 | 21 | 0.94 |
| AdaPlanner (Sun et al., 2024) | text-davinci-003 | 65 | 0.93 |
| Synapse (Zheng et al., 2024) | gpt-3.5-turbo | 100 | 0.98 |
| **SteP (Ours)** | text-davinci-003 | 10 | 0.96 |

Table 2: **Comparison to prior works** with success rates averaged across 45 MiniWoB++ tasks. SteP achieves competitive success rates while using significantly less data than prior works.

2024) uses ∼ 100 trajectories with 2-3 exemplars sampled from each of 48 tasks. In comparison, SteP uses 10 trajectories from only 6 tasks. SteP generalizes to remaining tasks through composition, i.e., by breaking them down into subtasks soluble by existing library policies. Thus, unlike prior work, SteP does not need to see exemplars from every task it is required to solve.

### 5.1.3 Why does a library of policies help over a single policy?

We observed that in prior works (Akter et al., 2023; Zhou et al., 2023), a common failure mode was an inability to navigate the website correctly to solve a complex, multi-step task. As a natural first solution, we add instructions and examples to the prompt to teach it how to solve such tasks. We created two baselines, Flat-4k and Flat-8k, containing such instructions and examples up to a context limit of 4000 and 8000 respectively.

In Table. 1, we see that Flat-4k improves upon prior work (0.15 → 0.20). However, when we go from Flat-4k to Flat-8k, the performance gains increase only marginally (0.20 → 0.23) even though we double context lengths.

On some intents, success rates even regress, e.g. on Shopping (0.2 → 0, 0.6 → 0.4), on Reddit (0.5 → 0.33, 0.6 → 0.4). This is because additional instructions create larger prompts, making it difficult for the model to pay attention.

SteP introduces a library of 14 policies, each with a small set of dedicated instructions and examples with prompt lengths under 4000 (see Appendix A.2). This

|  | Task | Flat Zero-shot suc↑ | Flat Zero-shot #act | Flat Few-shot suc↑ | Flat Few-shot #act | SteP Zero-shot suc↑ | SteP Zero-shot #act | SteP Few-shot suc↑ | SteP Few-shot #act |
|---|---|---|---|---|---|---|---|---|---|
| simple | click-option | 0.76 | 3.68 | 1.00 | 2.62 | 0.80 | 2.94 | 1.00 | 1.94 |
| | click-dialog-2 | 0.98 | 1.00 | 1.00 | 1.00 | 0.98 | 1.00 | 1.00 | 1.02 |
| | enter-date | 1.00 | 3.00 | 1.00 | 2.00 | 0.00 | 4.08 | 1.00 | 2.00 |
| | login-user | 0.96 | 3.42 | 1.00 | 3.00 | 1.00 | 3.06 | 1.00 | 3.00 |
| | grid-coordinate | 1.00 | 1.00 | 1.00 | 1.00 | 1.00 | 1.00 | 1.00 | 1.00 |
| complex | copy-paste-2 | 0.54 | 7.66 | 0.56 | 4.00 | 0.48 | 3.84 | 0.96 | 2.04 |
| | find-word | 0.22 | 2.62 | 0.26 | 5.18 | 0.12 | 2.92 | 0.98 | 2.00 |
| | choose-date-medium | 0.32 | 2.90 | 0.20 | 2.76 | 0.20 | 9.26 | 1.00 | 3.86 |
| | click-checkboxes-large | 0.00 | 8.40 | 0.20 | 8.40 | 0.00 | 7.00 | 1.00 | .20 |
| | click-checkboxes-transfer | 0.40 | 4.80 | 0.40 | 3.90 | 0.54 | 3.20 | 0.94 | 2.84 |
| | email-inbox | 0.40 | 7.00 | 0.70 | 4.50 | 0.00 | 3.00 | 0.90 | 5.20 |
| | simple-algebra | 0.14 | 8.80 | 0.30 | 6.78 | 0.04 | 4.38 | 0.74 | 2.00 |
| | login-user-popup | 0.46 | 6.28 | 0.46 | 3.52 | 0.46 | 5.82 | 1.00 | 4.88 |
| | search-engine | 0.38 | 3.64 | 0.38 | 3.16 | 0.26 | 4.46 | 1.00 | 4.30 |
| | book-flight | 0.00 | 16.00 | 0.10 | 11.10 | 0.00 | 13.52 | 0.90 | 9.14 |
| | **Mean (all 45 tasks)** | 0.60 | 3.70 | 0.72 | 3.38 | 0.65 | 3.43 | 0.96 | 2.89 |

Table 3: **Task-wise performance breakup** on MiniWoB++ on a subset of 15 tasks. See Appendix A.4 for a full breakup over 45 tasks.

results in smaller prompts that make fewer errors. Table 1 shows that SteP outperforms both Flat-4k (0.20 → 0.33) and Flat-8k (0.23 → 0.33). A key feature that helps SteP scale is that a single policy can cover multiple intents, e.g. search_order() covers 6 intents consisting of 30 tasks.

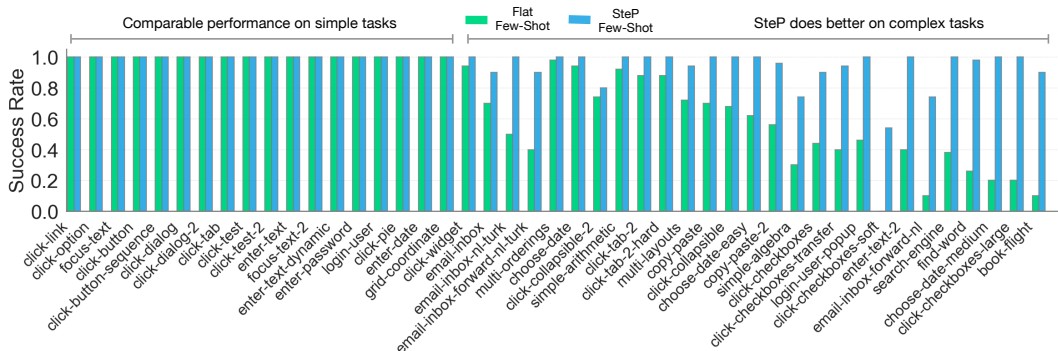

Figure 4: Success rate comparisons between `Flat Few-shot` and `SteP Few-shot` broken down over 45 MiniWob++ tasks (averaged over 50 seeds per task).

| Task | Metric | Flat Zero-shot | Flat Few-shot | SteP Zero-shot | SteP Few-shot | |
|------|--------|-----------|----------|-----------|----------|---|
| CANCEL FLIGHT | suc↑ | 0.10 | 1.00 | 0.20 | 1.00 | |
| | prog↑ | 0.15 | 1.00 | 0.80 | 1.00 | |
| | #act | 11.2 | 6.00 | 11.3 | 6.00 | |
| FIND BOOKING | suc↑ | 0.00 | 0.90 | 1.00 | 1.00 | |
| | prog↑ | 0.00 | 0.90 | 1.00 | 1.00 | |
| | #act | 11.0 | 4.10 | 3.00 | 3.00 | |
| SEARCH FLIGHT | suc↑ | 0.00 | 0.00 | 0.00 | 1.00 | |
| | prog↑ | 0.50 | 0.60 | 0.60 | 1.00 | |
| | #act | 11.0 | 11.0 | 11.0 | 5.00 | |
| UPDATE PASSENGER DETAILS | suc↑ | 0.00 | 0.50 | 0.30 | 0.65 | |
| | prog↑ | 0.00 | 0.90 | 0.60 | 0.90 | |
| | #act | 16.0 | 11.1 | 14.3 | 11.8 | |
| BOOK FLIGHT | suc↑ | 0.00 | 0.00 | 0.00 | 0.65 | |
| | prog↑ | 0.55 | 0.40 | 0.40 | 0.82 | |
| | #act | 26.0 | 26.0 | 25.7 | 22.2 | |
| MEAN | suc↑ | **0.02** | **0.48** | **0.30** | **0.86** | |
| | prog↑ | **0.24** | **0.76** | **0.68** | **0.94** | |
| | #act | 15.0 | 11.6 | 11.1 | 9.62 | |

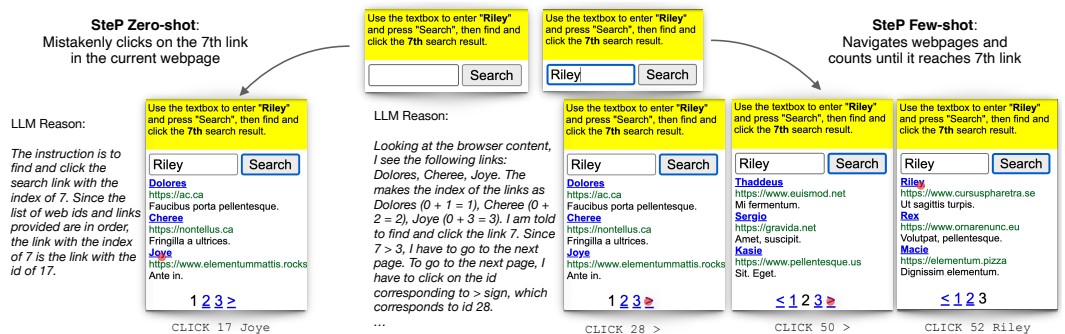

Figure 5: **(a)** Evaluation on 5 airline CRM tasks averaged over 20 randomized scenarios per task. **(b)** Simulator visualization of a book-flight task consisting of >20 steps. More details in Appendix C.

Figure 6: `SteP Few-shot` vs `SteP Zero-shot` on a search-engine task. The instruction asks to find the 7th link, however, it is ambiguous what 7 refers to. `SteP Few-shot` with an in-context example is able to ground the task in the UI and reason that the 7th link lies in the 2nd webpage.

On MiniWob++, in Table 3, we see a similar trend where `SteP` improves over `Flat Few-shot` (0.72 → 0.96). Fig. 4 shows comparisons across individual tasks. While performance is comparable on simpler tasks, as the task complexity increases `SteP` outperforms by greater margins. `SteP` breaks down complex tasks into smaller sub-tasks covered by policies in the library, e.g. `book_flight()` is broken down into `fill_text()`, `choose_date()`.

### 5.1.4   How context efficient is `SteP`?

A trade-off to `SteP` is that by introducing a library of policies, there is an overhead in passing control back and forth amongst policies. This leads to more number of calls to the model. However, since the prompt for each policy is significantly smaller, the total tokens ends up being smaller. For instance, when `SteP` solves a task it never has to see instructions and examples for policies not required in that task. Fig. 3(b) shows a histogram of context lengths for `SteP` and `Flat-8k` across all successful WebArena trajectories. `SteP` is distributed around a smaller number of tokens, averaging 22.7K compared to 52K. Hence total cost and inference time for `SteP` is lower than `Flat-8k`.

### 5.1.5   What is the effect of in-context examples?

We observe that while instructions are often sufficient to solve many web tasks, examples can provide a significant performance boost, particularly when these elements are stripped down lacking meaningful aria labels. On MiniWob++ Table 3, we see that few-shot examples provide performance gains for both `Flat` (0.60 → 0.72) and `SteP` (0.65 → 0.96).

We see a similar trend in AirlineCRM in Fig. 5 for both `Flat` (0.24 → 0.76) and `SteP` (0.68 → 0.94). Examples help associate language instructions with webpage elements, particularly for simplified pages when these are ambiguous. Fig. 6 shows a MiniWob++ search-engine task, where it is unclear what the 7th link implies. Fig. 7 shows `SteP` vs `Flat` with varying number of in-context examples on MiniWob++. We provide a maximum of 21 examples. We observe two main sources of improvement: (1) For the same number of examples (≤ 7), improvements come from decomposing task instructions into granular policy instructions (2) Each policy prompt contains dedicated in-context examples, allowing for more in-context examples (> 7) in each prompt.

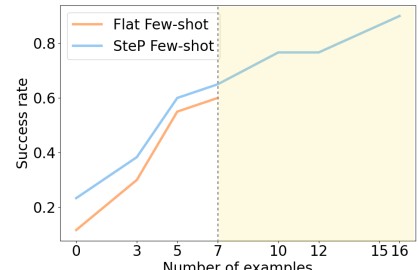

Figure 7: `SteP` vs `Flat` with varying in-context examples on subset of MiniWob++ tasks. Yellow region shows extra examples `SteP` packs in policies.

## 6   Limitations

We present `SteP` that dynamically composes policies to perform a diverse set of web tasks. Our results show that `SteP` outperforms both prior works and single policy baselines while being more context-efficient. However, `SteP` has several important limitations:

**(1) *Manually defined policies.*** For any new domain, the user needs to identify commonly occurring subtasks that are shared across multiple tasks. For each subtask, they then need to write a policy prompt that contains instructions on how to solve the subtask. Both identifying subtasks and writing policy prompts require domain knowledge. Once the policy prompts are written, however, SteP can automatically decide when to compose the appropriate policies from the library. The benefit of composition is that a small set of policies can cover a combinatorially large number of tasks. Automatically discovering useful policies from experience or demonstration data is an important direction of future work and would enable scaling to a large number of domains and websites automatically. **(2) *Communication overhead.*** By decomposing tasks into smaller policies, we also incur a communication overhead between policies that adds to inference times. One interesting solution would be to use smaller models for simpler policies and only escalate to larger models as needed. **(3) *Incomplete information.*** Finally, there are situations where a policy is unable to solve a subtask due to inadequate information being passed to it. Moreover, it fails to communicate what is missing to the parent policy, which can create an endless loop. An interesting future direction would be to explore how policies can share and update a common belief state to prevent such errors.

## Acknowledgements

We thank Kilian Weinberger, Michael Griffiths, Ramya Ramakrishnan, Adrian Botta, and the rest of ASAPP research and UI automation teams for their insightful feedback and suggestions. We also thank the anonymous reviewers for their constructive feedback, which greatly enhanced the quality of this paper.

## Ethics Statement

Equipping LLMs with the ability to carry out web tasks opens up a variety of possibilities for societal benefits and change. These range from reducing cognitive burden on humans of doing repetitive tasks to enabling greater accessibility for elderly and individuals with disabilities. However, we acknowledge the ethical implications that come with the use of such technologies, and list some of them below:

1. **Safety and Reliability.** LLMs automating web tasks raises concerns regarding misuse, including malicious automation aimed at spamming, phishing, or manipulating online systems. To mitigate these risks, it is crucial to implement stringent safeguards. These may include developing sophisticated detection algorithms to identify and block automated actions that exhibit patterns of misuse, ensuring that LLMs operate within ethical boundaries. Moreover, rigorous testing and validation protocols would ensure the technology's reliability and safety in open-world web environments.
2. **Privacy and Data Security.** LLMs interacting with web interfaces introduces potential risks such as unauthorized data access and privacy breaches. To mitigate these risks, it is crucial to place safeguards such as encrypting sensitive data, enforcing strict access controls, and adhering to strong privacy practices. Such transparent data handling policies are essential to maintain trust in these systems with end users.
3. **Employment Impact.** While automating web tasks with LLMs can enhance efficiency and accessibility, it is vital to consider the impact on employment. Care must be taken that the deployment of such technologies is to augment human capabilities rather than replace them, helping free them up for more creative and nuanced interactions. Addressing this requires holistic approaches, including policies to support workforce transition through re-skilling and up-skilling, and encouraging the development of new roles that leverage the unique strengths of human creativity.

## Reproducibility Statement

To promote reproducibility and transparency, we have taken several steps to ensure that `SteP` and our findings can be replicated and validated by the broader research community:

1. **Open source code.** We have linked the `SteP` implementation, including necessary code and documentation. We also include the prompts, the raw model predictions, and relevant noteboooks for greater reproducibility.
2. **Experimental Details.** Our paper includes detailed descriptions of the experiments along with additional details on hyperparamaters, prompts, environments included in the Appendix.
3. **Benchmarks and Models.** We clearly specify the web benchmarks and models used in our evaluations, including appropriate references and links to those. Since OpenAI models continue to evolve, we have included the raw model predictions so that the results are reproducible.
4. **Results and Ablations.** We present detailed results, including performance metrics, ablation studies, and comparisons with state-of-the-art methods. Our aim is to provide a clear and honest assessment of `SteP`'s capabilities and limitations.
5. **Limitations.** Acknowledging the importance of transparency in scientific communication, we discuss the limitations of our approach and discuss directions for future research.

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

# Appendix

## Table of Contents

# A Experiment Details

## A.1 Hyper-parameters

We use OpenAI API for model calls, `gpt-4-turbo-preview` as the default model for WebArena and `text-davinci-003` for other environments. We use a a temperature of 0.3 and set the number of calls to 3. Below are the exact API calls,

```
if (model_type == "gpt-4-turbo-preview"):
    response = openai.ChatCompletion.create(
        model=model_type,
        messages=[{"role": "user", "content": prompt}],
        temperature=0.3,
        top_p=1,
        n=3,
        max_tokens=max_tokens
    )
    response = response.choices[0].message.content.strip()

elif (model_type == "gpt-3.5-turbo"):
    response = openai.ChatCompletion.create(
        model=model_type,
        messages=[{"role": "user", "content": prompt}],
        temperature=0.3,
        top_p=1,
        n=3,
        max_tokens=max_tokens
    )
    response = response.choices[0].message.content.strip()

elif (model_type == "text-davinci-003"):
    response = openai.Completion.create(
        model=model_type,
        prompt=prompt,
        temperature=0.3,
        best_of=3,
        n=3,
        max_tokens=max_tokens
    )
    response = response.choices[0].text.strip()
```

Listing 1: Hyper-parameters for different models

## A.2 WebArena Policies and Prompts

We provide below the prompt template for WebArena that contains the list of policies, the base actions, examples for how to use the policies, and the base instruction template.

```
policies = """
Subroutine Actions:
`find_commits [query]`: Given you are in a project page, this Gitlab
subroutine searches for commits made to the project and retrieves
information about a commit. This function returns the answer to the query.

`search_issues [query]`: Given you are in my issue page, this Gitlab
subroutine searches issues that matches the query. Any objective
that says "open my latest issue" or "open issue with <keyword> in the
title" must be passed through this subroutine.

`create_project [query]`: Given you are in the create new project page,
this Gitlab subroutine completes the act of creating a project, adding members
    etc.
```

```
14
15  `create_group [query]`: Given you are in the create new group page,
16  this Gitlab subroutine completes the act of creating a group, adding members etc
        .
17
18  `find_subreddit [query]`: This Reddit subroutine finds a subreddit corresponding
19  to the query. The query can either be the name of the subreddit or a vague
20  description of what the subreddit may contain. The subroutine hands back
21  control once it navigates to the subreddit.
22
23  `find_user [user_name]`: This Reddit subroutine navigates to the page of a user
24  with user_name. The page contains all the posts made by the user.
25
26  `find_customer_review [query]`: This CMS subroutine finds customer reviews for
27  a particular product using the query to specify the kind of review.
28
29  `find_order [query]`: This CMS subroutine finds an order corresponding to a
30  particular customer or order number.
31
32  `search_customer [query]`: This CMS subroutine finds a customer given some
33  details about them such as their phone number.
34
35  `search_order [question]`: This Shopping subroutine searches orders to answer
36  a question about my orders
37
38  `list_products [query]`: This Shopping subroutine find products that match a
        query
39
40  `search_reviews [query]`: This Shopping subroutine searches reviews to answer
41  a question about reviews
42
43  `find_directions [query]`: This Maps subroutine finds directions between two
44  locations to answer the query
45
46  `search_nearest_place [query]`: This Maps subroutine find places near a given
        location
47  """
48
49
50  example_actions = """
51  click [7]
52
53  type [15] [Carnegie Mellon University] [1]
54
55  stop [Closed]
56
57  hover [15]
58
59  scroll [down]
60
61  note [Spent $10 on 4/1/2024]
62
63  find_commits [How many commits did user make to diffusionProject on 03/23/2023?]
64
65  search_issues [Open my latest updated issue that has keyword "better"
66  in its title to check if it is closed]
67
68  create_project [Create a new public project "awesome-llms" and add primer,
69  convexegg, abishek as members]
70
71  create_group [Create a new group "coding_friends" with members qhduan, Agnes-U]
72
73  find_subreddit [books]
74
75  find_user [AdamCannon]
```

```
76
77  find_customer_review [Show me customer reviews for Zoe products]
78
79  find_order [Most recent pending order by Sarah Miller]
80
81  search_customer [Search customer with phone number 8015551212]
82
83  search_order [How much I spend on 4/19/2023 on shopping at One Stop Market?]
84
85  list_products [List products from PS4 accessories category by ascending price]
86
87  search_reviews [List out reviewers, if exist, who mention about ear cups being
        small]
88
89  find_directions [Check if the social security administration in Pittsburgh
90  can be reached in one hour by car from Carnegie Mellon University]
91
92  search_nearest_place [Tell me the closest cafe(s) to CMU Hunt library]
93  """
94
95
96  base_actions = """
97  Page Operation Actions:
98  `click [id]`: This action clicks on an element with a specific
99  id on the webpage.
100 `type [id] [content] [press_enter_after=0|1]`: Use this to type the
101 content into the field with id. By default, the "Enter" key is pressed
102 after typing unless press_enter_after is set to 0.
103 `hover [id]`: Hover over an element with id.
104 `press [key_comb]`:  Simulates the pressing of a key combination
105 on the keyboard (e.g., Ctrl+v).
106 `scroll [direction=down|up]`: Scroll the page up or down.
107 `note [content]`: Use this to make a personal note of some content
108 you would like to remember. This shows up in your history of previous
109 actions so you can refer to it.
110 `go_back`: Navigate to the previously viewed page.
111
112 general_instruction_template = """
113 You are an AI assistant performing tasks on a web browser.
114 To solve these tasks, you will issue specific actions.
115
116 The actions you can perform fall into several categories:
117 {base_actions}
118
119 {policies}
120
121 {example_actions}
122
123 You will be provided with the following,
124     OBJECTIVE:
125     The goal you need to achieve.
126     OBSERVATION:
127     A simplified text description of the current browser
128     content, without formatting elements.
129     URL:
130     The current webpage URL
131     PREVIOUS ACTIONS:
132     A list of your past actions with an optional response,
133     e.g. 1 = find_commits [query]
134
135 You need to generate a response in the following format.
136 Please issue only a single action at a time.
137   REASON:
138   Your reason for selecting the action below
139   ACTION:
```

```
140     Your action
141   """
142
143
144
145 Tab Management Actions:
146 `new_tab`: Open a new, empty browser tab.
147 `tab_focus [tab_index]`: Switch the browser's focus to a specific
148 tab using its index.
149 `close_tab`: Close the currently active tab.
150
151 URL Navigation Actions:
152 `goto [url]`: Navigate to a specific URL.
153 `go_back`: Navigate to the previously viewed page.
154 `go_forward`: Navigate to the next page (if a previous 'go_back'
155 action was performed).
156
157 Completion Action:
158 `stop [answer]`: Issue this action when you believe the task is
159 complete. If the objective is to find a text-based answer,
160 provide the answer in the bracket.
161   """
```

We provide the prompts for all 14 commonly used policies below and for the reset, we refer the reader to the code base.

```
1  find_commits = {
2  "instruction": """
3  {general_instruction_template}
4
5  Please follow these general instructions:
6  * To find a list of all commits, you must navigate to the commits section of the
        repository
7  * Look at the first and last date in your observation to know if the desired
        date is in the range
8  * If it's in the range but not visible, that means no commits were made on that
        date
9  * If the date is outside of the range, you need to scroll up/down to get to the
        desired date range. Scrolling down takes you to a date earlier in time (e.g
        . Feb 2023 is earlier in time than Mar 2023)
10 * To count commits from a specific author, count the number of times their
        avatar (e.g. img "<author> avatar") appears in the observation.
11  """,
12
13  "examples": [...]
14  }
15
16  search_issues = {
17  "instruction": """
18  {general_instruction_template}
19
20  Please follow these general instructions:
21  * First navigate the Issues page
22  * Once you are in the Issues page, you MUST first navigate to all issues so that
         you see both open and closed issues for solving the objective
23  * You may not see all issues listed at once, use the search bar to search for
        appropriate keywords and filter down to relevant set of issues
24  * If the objective says to "Open ... issue, check if it is X", you must first
        open the specific issue page by clicking it. Do not stop [] until you have
        navigated to the specific issue page.
25  * Once you are on the issue page, return the appropriate status
26  * In your status, if the objective is to check if an issue is open or closed,
        respond as though you are answering a question, e.g. "No, it is open", "Yes
        , it is closed
```

```
27 """,
28
29 "examples": [...]
30 }
31
32 create_project = {
33 "instruction": """
34 {general_instruction_template}
35
36 Please follow these general instructions:
37 * To add new members, once you have created the project, click on Project
       Information in the sidebar to be guided to a link with memmbers.
38 * When adding members, first type their name, then click on their name from the
       down down. Consult PREVIOUS ACTIONS to see if you have typed and selected
       the names.
39 """,
40
41 "examples": [...]
42 }
43
44 create_group = {
45 "instruction": """
46 {general_instruction_template}
47
48 Please follow these general instructions:
49 * To add new members, click on the Members tab in the side pane. If you don't
       see it, click on Group Information in the sidebar to be guided to a link
       with memmbers.
50 * When adding members, first type their name, then click on their name from the
       down down. Consult PREVIOUS ACTIONS to see if you have typed and selected
       the names.
51 """,
52
53 "examples": [...]
54 }
55
56 find_subreddit = {
57 "instruction": """
58 {general_instruction_template}
59
60 Please follow these instructions to solve the subtask:
61 * The objective find_subreddit [query] asks you to navigate to the subreddit
       that best matches the query. The query can be specific or vague.
62 * The first step is to navigate to Forums to see the list of subreddits. However
       , if you have done this already (indicated as non empty PREVIOUS ACTIONS),
       do not repeat this step.
63 * Under forums, you will see only a subset of subreddits. To get the full list
       of subreddits, you need to navigate to the Alphabetical option.
64 * To know you can see the full list of subreddits, you will see 'All Forums' in
       the observation
65 * Often you will not find a focused subreddit that exactly matches your query.
       In that case, go ahead with the closest relevant subreddit.
66 * To know that you have reached a subreddit successfully, you will see '/f/
       subreddit_name' in the observation.
67 * Once you have navigated to any specific subreddit, return stop [N/A]. Even if
       the subreddit is generally related and not specific to your quwey, stop
       here and do not try to search again for another subreddit.
68 """,
69
70 "examples": [...]
71 }
72
73 find_user = {
74 "instruction": """
75 {general_instruction_template}
```

```
76
77  Please follow these instructions to solve the subtask:
78  * The objective find_user [user_name] asks you to navigate the page of a user
        with user_name
79  * To do so, look at the current base URL (e.g. https://webarena-env-reddit.
        awsdev.asapp.com) and add a suffix /user/user_name, i.e.
80  goto [https://webarena-env-reddit.awsdev.asapp.com/user/user_name]
81  * Once you have navigated to the user page (as seen in your past actions),
        return stop [N/A]
82  """,
83
84  "examples": [...]
85  }
86
87  find_customer_review = {
88  "instruction": """
89  {general_instruction_template}
90
91  Please follow these instructions to solve the subtask:
92  * The objective find_customer_review [query] asks you to navigate to the product
        page containing customer reviews.
93  * To navigate to a review, first click on REPORTS in the side panel
94  * Once you have clicked on REPORTS, and you see the Reports panel with Marketing
        , Sales, Reviews, Customers etc, click on By Products under Customers.
95  * Once you are in the Product Reviews Report, you need to locate the product by
        searching for it. Use the gridcell below Product to search for a product.
        Do not use other search boxes. Look at the example below where I show you
        how to search for Zoe in the correct gridcell.
96  * When searching for a product, search the first word only like Zoe, or Antonia
        or Chloe.
97  * Once the product shows up, click on 'Show Reviews'.
98  * Once all the reviews show up, return stop [N/A] to hand back control to the
        agent that queried you.
99  """,
100 "examples": [...]
101 }
102
103 find_order = {
104 "instruction": """
105 {general_instruction_template}
106
107 Please follow these instructions to solve the subtask:
108 * The objective find_order [query] asks you to navigate to the order page
        corresponding to the query
109 * To navigate to orders, first go to SALES in the side panel
110 * Once you have clicked on SALES, go to Orders
111 * Once you are in the orders page, you have to use the 'Filter' button to filter
        down to desired criteria
112 * Desired criterias include filtering down to a specific order ID field or Name
        field. ONLY use fields that are in the objective
113 * You MUST use Filter to find orders instead of using the search bar
114 * If there are any active filters, be sure to clear them before entering your
        filter criteria
115 * In your filtered list of orders, if you don't find the desired order, make
        sure to scroll down till you find the order or reach end of page (typically
        indicated by 'Copyright ...' in the observation)
116 * Once you have found the order, go to View to open the order
117 * Once you are in the desired order page (as noted by "Order & Account
        Information") you MUST return stop [N/A] to hand back control to the agent
        that queried you. Do not go back to another page.
118 """,
119 "examples": [...]
120 }
121
122 search_customer = {
```

```
123  "instruction": """
124  {general_instruction_template}
125
126  Please follow these instructions to solve the subtask:
127  * The objective search_customer [query] asks you to search for customer details
          corresponding to the query
128  * To navigate to customers, first click on CUSTOMERS in the side panel
129  * Once you have clicked on CUSTOMERS, click on All Customers.
130  * Once you are in the customers page, you have to use the 'Search by keyword'
          text box to search for your customer. Always be sure to search first. For
          example, for find_order [Search customer with phone number 8015551212],
          search 8015551212.
131  * If the page shows a number has already been searched, click on Clear All first
          . Then proceed with the search.
132  * Once you are done with the search, and the customer with matching query shows
          up, you MUST return stop [N/A] to hand back control to the agent that
          queried you. Do not go back to another page.
133  """,
134  "examples": [...]
135  }
136
137  search_order = {
138  "instruction": """
139  {general_instruction_template}
140
141  Please follow these GENERAL INSTRUCTIONS:
142  * Navigate to My Account, then My Orders to access all orders.
143  * The orders are sorted by descending date. Click on Page Next to go to a
          earlier date. Click on Page Previous to go to a earlier date.
144  If you don't see an order for a date, and the first order on the page is after
          the date, the last order on the page is before the date, then it means
          there is no order for the date.
145  * In your REASON, state what is the current range, what range you are looking
          for, and whether you should search for an earlier or a later date.
146  * If you have to find the total amount you spent on orders that span multiple
          pages, use note [Spent $10 on 4/1/2024] to make a personal note before
          moving on to the next page. When you are done, you can look at PREVIOUS
          ACTIONS to find all notes.
147  * When you are adding numbers, work out each addition step by step in REASON.
148  * Use go_back to go back to a previous page from an order. But before you do,
          use note [] to make a note that you checked the page, e.g. note [Checked
          order on 11/29/2023, no picture frame.]
149  * If you are in an order page and need to go back, issue go_back. Don't click on
           My Orders else you have to start from all over again.
150  * Do not keep visiting the same order page over and over again!
151  To prevent this, whenever you visit a page, always make a note. For example note
          [Nothing relevant purchased on September 29, 2022]
152  See note [] to see what dates you have visit, and be sure to not visit that page
          again.
153  * Once you are done visiting all the pages, return stop [answer] with the answer
           to the query.
154  * If the question is how much did I spend on a date, and I didn't spend anything
          , return stop [$0]
155  * If the status of an order shows cancelled, that means I did not spend that
          money
156  * If you are asked to change the delivery address on an order, you can't. Reply
          stop [N/A]
157  """,
158
159  "examples": [...]
160  }
161
162  list_products = {
163  "instruction": """
164  {general_instruction_template}
```

```
165
166 Please follow these instructions to solve the subtask:
167 * To find a product category, you MUST use hover [id] to expand the popup Menu
        till you find the leaf element that has no popup. Then click [id] on the
        leaf element.
168 For exmaple, to find PS 4 accessories you must hover over Video Games, then
        hover over Playstation 4, then click on Accessories.
169 Use note [] eveytime you hover on an item, and don't find the category. This is
        to ensure you don't keep trying the same category repeatedly.
170 * To sort current list of products by price and in ascending order, you MUST use
         the goto [url] action by appending ?product_list_order=price to the
        current URL. For example:
171 * To sort in descending order, you MUST use the goto [url] action by appending ?
        product_list_order=price&product_list_dir=desc, e.g.
172 * To list all items less than a particular price, e.g. $25, you MUST use the
        goto [url] action by appending ?price=0-25
173 * Once you are done in stop [N/A]
174 * If the OBJECTIVE asks you to show the most expensive product, you must click
        on the product.
175 """,
176
177 "examples": [...]
178 }
179
180 search_reviews = {
181 "instruction": """
182 {general_instruction_template}
183
184 Please follow these instructions to solve the subtask:
185 * If you are not in the product page, you can search for the product using the
        search bar.
186 * To find the list of reviews, search for a link with Reviewers. If you can't
        find it, scroll down to search for it.
187 * Iterate over all reviews. For every relevant review, make a note [
        reviewer_name: review_info]. Record the relevant reviewer_name and review
        VERBATIM. Once you are done with all the reviews in a page, scroll down to
        access more reviews.
188 * Refer to PREVIOUS ACTIONS to know which reviews you have noted already. If you
         have noted a review already, look for the next review in your current
        OBSERVATION or scroll down.
189 * Do NOT repeat the note [] action for the same review.
190 * Not all reviews will be visible on the reviews page. You MUST scroll down till
         you reach the end of the page. You will know that you have reached the end
         of the page if you see Contact Us in the OBSERVATION.
191 * Once you have scrolled through all reviews, combine all your noted reviews
        that you can find under PREVIOUS ACTIONS. To combine, create a list of
        dicts where every dict has a name and review key. Be sure to capture ALL
        the reviews in your note. Return that as stop [{name: reviewer_name_1,
        review: review_1}, {name: reviewer_name_2, review: review_2}, ..]
192 """,
193
194 "examples": [...]
195 }
196
197 find_directions = {
198 "instruction": """
199 {general_instruction_template}
200
201 Please follow these instructions to solve the subtask:
202 * First click on "Find directions between two points", then enter From and To
        Fields, and click search.
203 * If you have to find directions to social security administration in Pittsburgh
        , search for it in a structured format like Social Security Administration,
         Pittsburgh.
204 """,
```

```
205
206 "examples": [...]
207 }
208
209 search_nearest_place = {
210 "instruction": """
211 {general_instruction_template}
212
213 Please follow these instructions to solve the subtask:
214 * For searches that refer to CMU, e.g.  "find cafes near CMU Hunt Library"
215 a. You have to first center your map around a location. If you have to find
        cafes near CMU Hunt Library, the first step is to make sure the map is
        centered around Carnegie Mellon University. To do that, first search for
        Carnegie Mellon University and then click [] on a list of location that
        appears. You MUST click on the Carnegie Mellon University location to
        center the map. Else the map will not centered. E.g click [646]
216 b. Now that your map is centered around Carnegie Mellon University, directly
        search for "cafes near Hunt Library". Do not include the word CMU in the
        search item.
217 The word CMU cannot be parsed by maps and will result in an invalid search.
218 c. When your search returns a list of elements, return them in a structured
        format like stop [A, B, C]
219 * For searches that don't refer to CMU
220 a. No need to center the map. Directly search what is specified in OBJECTIVE, e.
        g. "bars near Carnegie Music Hall"
221 b. When your search returns a list of elements, return them in a structured
        format like stop [A, B, C]
222 * Be sure to double check whether the OBJECTIVE has CMU or not and then choose
        between instruction 1 and 2.
223 * Remember that the word CMU cannot be typed in the search bar as it cannot be
        parsed by maps.
224 * Remember that if you want to center your map around Carnegie Mellon University
        , you have to click on it after you search for it. Check your PREVIOUS
        ACTIONS to confirm you have done so, e.g. click [646] should be in the
        previous actions.
225 """,
226
227 "examples": [...]
228 }
```

## A.3  MiniWob++ Policies

On MiniWob++, we identified a set of 7 policies that solve commonly recurring subtasks, e.g. filling different text boxes, choosing dates from a date picker, processing emails, etc. For each of these policies, we collect a few in-context examples to cover the subtask. At test time, SteP composes these policies to solve a wide number of tasks. Table 4 below shows the various policies and tasks they cover.

## A.4  Task-wise Performance on MiniWoB++

Table 5 shows a task-wise performance breakup on the MiniWoB++ benchmark for various models. We choose a set of 45 tasks that don't require visual reasoning. We run 50 seeds per task and report the average success rates and number of actions. We divide tasks into simple and complex, where complex tasks might require multiple policies to solve the task. While on simple tasks almost all methods are equivalent, on complex tasks SteP Few-shot outperforms Flat Few-shot on every task.

Fig. 8 shows an example of SteP for a book-flight task. Book-flight is a relatively complex task, requiring filling in drop down text boxes and filling dates. SteP outperforms Flat (0.10 → 0.90), where both methods have access to a single demonstration of book flight. SteP composes policies FILL_TEXT() to fill in the Flight-from and Flight-to box, and then

| Method | Examples | Tasks covered by examples |
|---|---|---|
| Flat | 7 | choose-date, book-flight |
| SteP | 21 | |
| |— WEB_AGENT | 3 | |
| |— FILL_TEXT | 5 | choose-date, book-flight |
| |— CHOOSE_DATE | 4 | search-engine, click-tab-2 |
| |— SEARCH_LINK | 3 | click-checkbox, email- |
| |— SEARCH_TAB | 1 | inbox |
| |— CLICK_CHECKBOX | 2 | |
| |— PROCESS_EMAIL | 3 | |

Table 4: On MiniWob++, library of policies used by SteP, number of in-context examples per policy and the tasks covered. Each example is a state-action pair at particular timestep. Each policy in SteP requires fewer in-context examples compared to Flat, but combined together they cover many more tasks.

CHOOSE_DATE() policies to pick a date from the datepicker. Each of these policies are more robust than Flat at solving the subproblems, resulting in a higher overall success rate.

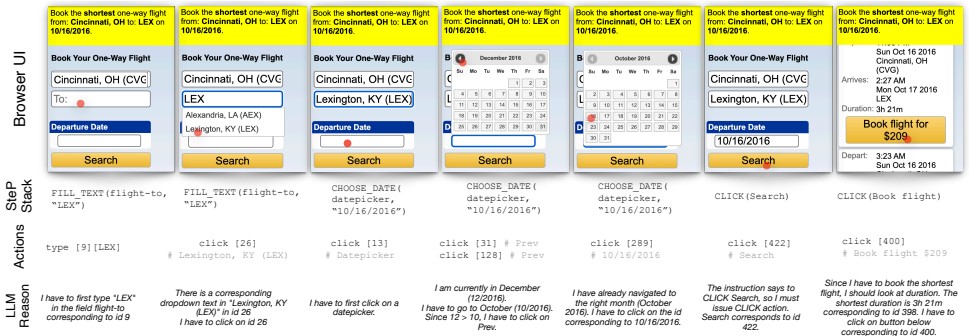

Figure 8: Outputs from SteP Few-shot on book-flight task showing hierarchical task planner actions, low-level web policy actions, and LLM reasoning.

# B  Ablations

## B.1  Effect of Chain-of-Thought Reasoning

While we initially did not have chain-of-thought reasoning [2], we found that adding it in helped SteP Few-shot rationalize a particular action well and boost performance. We wanted to understand which tasks in particular were helped by chain-of-thought reasoning, and what the trade-offs were.

We wanted to test the following hypothesis for SteP Few-shot with and without chain-of-thought:

1. **Hypothesis 1: Chain-of-thought reasoning helps across all tasks.** Even though chain-of-thought reasoning makes prompts slightly longer, the added step of rationalizing actions should always boost performance.

2. **Hypothesis 2: Chain-of-thought reasoning particularly helps in multi-step tasks.** Multi-step tasks often require breaking down a problem into a set of steps and executing each step. While demonstrations certainly show how to break down task, adding chain-of-thought better rationalizes this breakdown and helps generalize to new tasks not covered in the demonstrations.

---

[2]Chain-of-Thought Prompting Elicits Reasoning in Large Language Models https://arxiv.org/abs/2201.11903

| | Task | Flat Zero-shot | | Flat Few-shot | | SteP Zero-shot | | SteP Few-shot | |
|---|---|---|---|---|---|---|---|---|---|
| | | %suc↑ | #act↓ | %suc↑ | #act↓ | %suc↑ | #act↓ | %suc↑ | #act↓ |
| simple | click-link | 0.94 | 1.00 | 1.00 | 1.00 | 1.00 | 1.00 | 1.00 | 1.00 |
| | click-option | 0.76 | 3.68 | 1.00 | 2.62 | 0.80 | 2.94 | 1.00 | 1.94 |
| | focus-text | 1.00 | 1.00 | 1.00 | 1.00 | 1.00 | 1.00 | 1.00 | 1.00 |
| | click-button | 0.98 | 1.02 | 1.00 | 1.00 | 1.00 | 1.00 | 1.00 | 1.00 |
| | click-button-sequence | 0.96 | 2.00 | 1.00 | 2.00 | 1.00 | 2.00 | 1.00 | 2.00 |
| | click-dialog | 1.00 | 1.06 | 1.00 | 1.20 | 1.00 | 1.28 | 1.00 | 1.02 |
| | click-dialog-2 | 0.98 | 1.00 | 1.00 | 1.00 | 0.98 | 1.00 | 1.00 | 1.02 |
| | click-tab | 1.00 | 1.00 | 1.00 | 1.00 | 0.98 | 1.00 | 1.00 | 1.04 |
| | click-test | 1.00 | 1.00 | 1.00 | 1.00 | 1.00 | 1.00 | 1.00 | 1.00 |
| | click-test-2 | 1.00 | 1.00 | 1.00 | 1.00 | 1.00 | 1.00 | 1.00 | 1.00 |
| | enter-text | 1.00 | 2.50 | 1.00 | 2.00 | 1.00 | 2.10 | 1.00 | 2.00 |
| | focus-text-2 | 1.00 | 1.00 | 1.00 | 1.00 | 1.00 | 1.00 | 1.00 | 1.00 |
| | enter-text-dynamic | 0.98 | 2.44 | 1.00 | 2.00 | 0.98 | 2.06 | 1.00 | 2.00 |
| | enter-password | 1.00 | 3.08 | 1.00 | 3.00 | 1.00 | 3.20 | 1.00 | 4.52 |
| | login-user | 0.96 | 3.42 | 1.00 | 3.00 | 1.00 | 3.06 | 1.00 | 3.00 |
| | click-pie | 1.00 | 3.00 | 1.00 | 3.00 | 1.00 | 3.00 | 1.00 | 3.00 |
| | enter-date | 1.00 | 3.00 | 1.00 | 2.00 | 0.00 | 4.08 | 1.00 | 2.00 |
| | grid-coordinate | 1.00 | 1.00 | 1.00 | 1.00 | 1.00 | 1.00 | 1.00 | 1.00 |
| | click-widget | 0.94 | 1.00 | 0.94 | 1.00 | 0.94 | 1.00 | 1.00 | 1.00 |
| complex | email-inbox | 0.40 | 7.00 | 0.70 | 4.50 | 0.00 | 3.00 | 0.90 | 5.20 |
| | email-inbox-nl-turk | 0.40 | 7.20 | 0.50 | 6.00 | 0.00 | 2.90 | 1.00 | 4.58 |
| | email-inbox-forward-nl-turk | 0.30 | 5.08 | 0.40 | 4.80 | 0.00 | 3.50 | 0.90 | 4.30 |
| | multi-orderings | 0.56 | 3.60 | 0.98 | 3.98 | 0.76 | 4.28 | 1.00 | 4.00 |
| | choose-date | 0.20 | 3.00 | 0.94 | 3.60 | 0.20 | 5.80 | 1.00 | 5.40 |
| | click-collapsible-2 | 0.60 | 3.64 | 0.74 | 4.14 | 0.34 | 3.34 | 0.80 | 4.50 |
| | simple-arithmetic | 0.82 | 2.12 | 0.92 | 2.12 | 0.54 | 2.66 | 1.00 | 2.00 |
| | click-tab-2 | 0.76 | 5.58 | 0.88 | 4.62 | 0.88 | 2.84 | 1.00 | 2.24 |
| | click-tab-2-hard | 0.68 | 3.36 | 0.88 | 3.84 | 0.76 | 3.06 | 1.00 | 2.42 |
| | multi-layouts | 0.42 | 4.46 | 0.72 | 3.94 | 0.66 | 4.82 | 0.94 | 4.00 |
| | copy-paste | 0.14 | 2.14 | 0.70 | 3.48 | 0.98 | 2.72 | 1.00 | 2.00 |
| | click-collapsible | 0.54 | 1.76 | 0.68 | 1.88 | 0.86 | 1.88 | 1.00 | 2.00 |
| | choose-date-easy | 0.74 | 2.74 | 0.62 | 2.62 | 0.20 | 10.18 | 1.00 | 3.10 |
| | copy-paste-2 | 0.54 | 7.66 | 0.56 | 4.00 | 0.48 | 3.84 | 0.96 | 2.04 |
| | simple-algebra | 0.14 | 8.80 | 0.30 | 6.78 | 0.04 | 4.38 | 0.74 | 2.00 |
| | click-checkboxes | 0.40 | 4.90 | 0.44 | 5.94 | 0.74 | 3.20 | 0.90 | 3.14 |
| | click-checkboxes-transfer | 0.40 | 4.80 | 0.40 | 3.90 | 0.54 | 3.20 | 0.94 | 2.84 |
| | login-user-popup | 0.46 | 6.28 | 0.46 | 3.52 | 0.46 | 5.82 | 1.00 | 4.88 |
| | click-checkboxes-soft | 0.00 | 7.00 | 0.00 | 7.30 | 0.04 | 6.94 | 0.54 | 5.64 |
| | enter-text-2 | 0.00 | 2.60 | 0.40 | 5.20 | 0.40 | 2.00 | 1.00 | 2.00 |
| | email-inbox-forward-nl | 0.10 | 5.04 | 0.10 | 4.58 | 0.00 | 3.24 | 0.74 | 4.74 |
| | search-engine | 0.38 | 3.64 | 0.38 | 3.16 | 0.26 | 4.46 | 1.00 | 4.30 |
| | find-word | 0.22 | 2.62 | 0.26 | 5.18 | 0.12 | 2.92 | 0.98 | 2.00 |
| | choose-date-medium | 0.32 | 2.90 | 0.20 | 2.76 | 0.20 | 9.26 | 1.00 | 3.86 |
| | click-checkboxes-large | 0.00 | 8.40 | 0.20 | 8.40 | 0.00 | 7.00 | 1.00 | 6.20 |
| | book-flight | 0.00 | 16.00 | 0.10 | 11.10 | 0.00 | 13.52 | 0.90 | 9.14 |
| | Mean | **0.60** | 3.70 | **0.72** | 3.38 | **0.65** | 3.43 | **0.96** | 2.89 |

Table 5: Task-wise performance breakup on MiniWoB++ benchmark on a set of 45 tasks.

We compared `SteP Few-shot` with two versions - having chain of thought, and not having. Fig. 9 shows a plot of success rate for each of the 3 clusters of tasks - single, composite, multi.

**Hypothesis 1: Chain-of-thought reasoning helps across all tasks..** We find this to be true since for all tasks, chain-of-thought performs either equally or better. This confirms that the extra tokens consumed by the reasoning does not hurt performance and in fact helps significantly in some cases.

**Hypothesis 2: Chain-of-thought reasoning particularly helps in multi-step tasks.** We find this to be true as well. Looking at the multi-step tasks, chain-of-thought has the largest performance gains compared to any other cluster. The performance gains are the largest in book-flight and search-engine where the horizon length is the largest. In comparison, for single and composite the performance gains vary, being higher for certain tasks like choose-date and find-word and zero for others like click tasks.

## B.2 Model Scaling

In MiniWob++, while we developed the prompts with `text-davinci-003`, we wanted to compare how performance varies with newer models `gpt-3.5-turbo` and `gpt-4`. `gpt-3.5-turbo` is also an InstructGPT model optimized for chat and trained at 1/10th the price of `text-davinci-003`. `gpt-4` is a significantly larger model, and capable of solving more complex tasks.

We wanted to test the following hypothesis:

1. **Hypothesis 1: GPT-4 improves performance across all methods, but both decomposition and few-shot examples help.** GPT-4 is a powerful model and with an exhaustive set of instructions in the prompt, it should be able to achieve perfect performance. However, designing such exhaustive prompts for all web tasks is challenging. We hypothesize that decomposition helps break down and scope instructions leading to better performance for GPT-4. Similarly, few-shot examples helps GPT-4 ground instructions in the webpage.

2. **Hypothesis 2: `gpt-3.5-turbo` slightly worse than `text-davinci-003` given few-shot examples.** Practioners have noted that while `gpt-3.5-turbo` has better 0 shot performance, `text-davinci-003` is trained on a more diverse set of tasks and performs better with k-shot learning https://scale.com/blog/chatgpt-vs-davinci#Introduction. Since 0 shot performance for web-

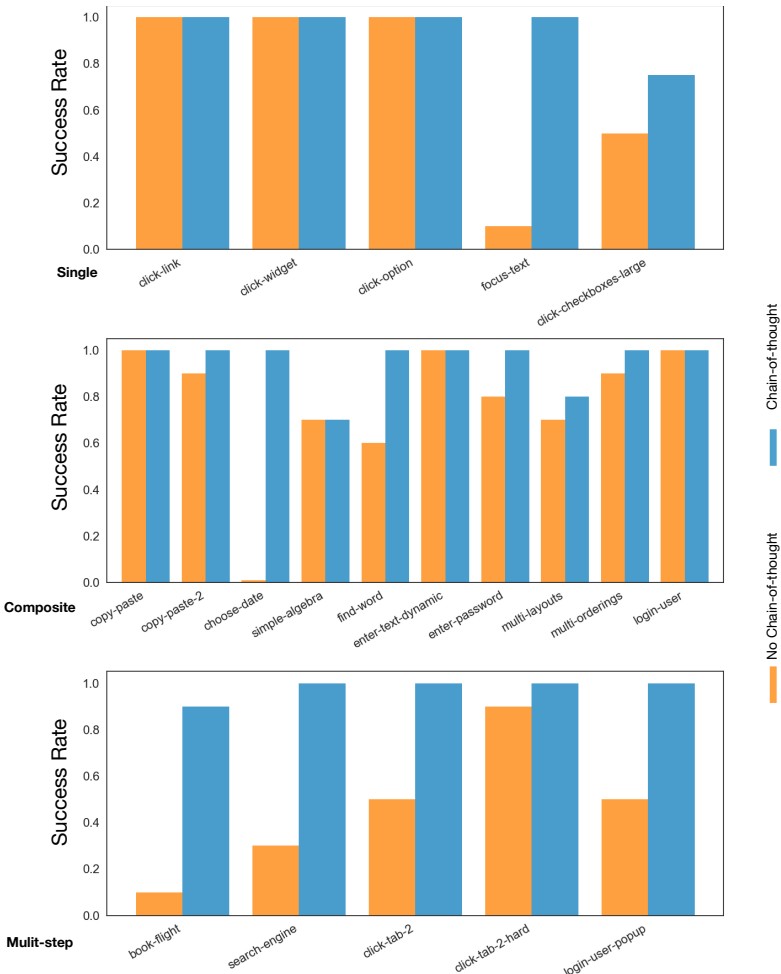

Figure 9: Success rate of `SteP Few-shot` with both chain-of-thought and without.

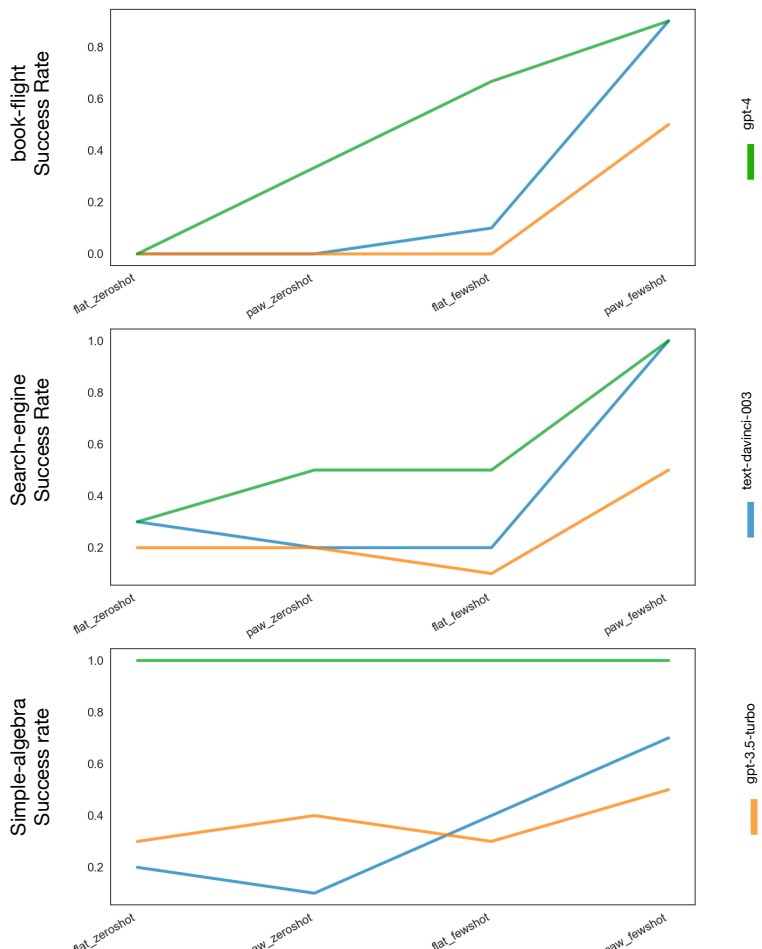

Figure 10: Success rate of all methods with different models.

tasks is challenging without exhaustive instructions, we hypothesize that `text-davinci-003` will perform better.

We compare three language models `text-davinci-003`, `gpt-3.5-turbo` and `gpt-4` for all baselines on 3 tasks from MiniWoB++ - book-flight, search-engine, simple-algebra. Fig. 10 shows a plot for each of these tasks.

**Hypothesis 1: GPT-4 improves performance across all methods, but both decomposition and few-shot examples help.** We find this to be true. GPT-4 is unambiguously better than any other model and improves all methods. However, as shown in book-flights and search-engine, both decomposition (`SteP`) and few-shot examples boost performance. In simpler reasoning tasks like simple algebra, it gets it correct right away.

**Hypothesis 2: `gpt-3.5-turbo` slightly worse than `text-davinci-003` given few-shot examples.** We also find evidence for this. `text-davinci-003` with few-shot examples always outperforms `gpt-3.5-turbo`. Although, in simple-algebra, zero shot performance of `gpt-3.5-turbo` is better than `text-davinci-003`, matching what other practitioners have observed in other tasks.

### B.3 Evaluation with Llama model

We also compare `Flat` and `SteP` with open-source models like LLaMA-{7B, 13B}. The models are all pre-trained instruction following models without any additional fine-tuning.

The prompts used are the same as `Flat Zero-shot` and `SteP Zero-shot`. Overall, we find the performance to be lower than gpt-{3,3.5,4}. The drop in performance could be due to a number of reasons such as the model size or the training data on which the models are trained. However, we find that `SteP` still outperforms `Flat` for many tasks, which we discuss below.

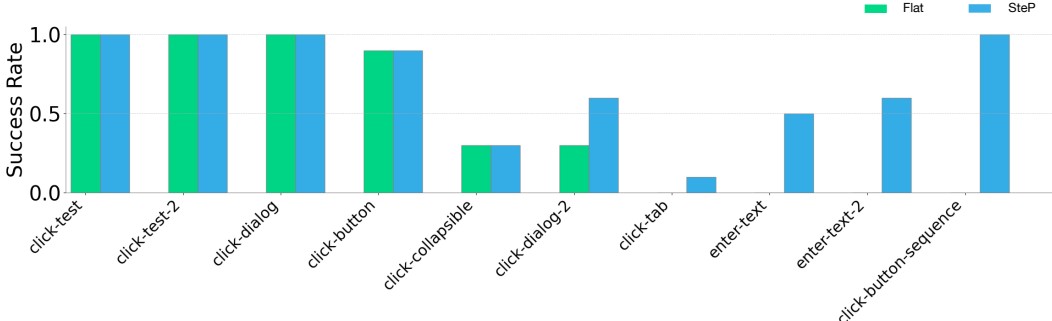

Figure 11: Evaluation with LlaMA-13B on a subset of MiniWob++ tasks. Success rate breakdown of 10 tasks averaged over 10 seeds over task.

Figure. 11 shows a comparison of `Flat` and `SteP` on a subset of MiniWob++ tasks. We see that `Flat` matches `SteP` for simple CLICK tasks, it fails for slightly more complex tasks that require multiple actions, e.g. multiple CLICK or TYPE.

We cluster common failure modes for both `Flat`, `SteP` below:

**1. Repeating the same action in multi-step tasks.** For tasks that require a sequence of actions, `Flat` tends to get stuck in the first step repeating the same action over and over. It fails to interpret the previous actions or changes to the browser content to update the predicted actions.

For example, in the `click-button-sequence` task, for the following input

```
CONTEXT:
Click button ONE, then click button TWO.
BROWSER CONTENT:
<body id=1 val= />
<div id=2 val=wrap />
<div id=3 val=area />
<button id=4 val=ONE />
<button id=5 val=TWO />
PREVIOUS ACTIONS:
CLICK 4
```

the model produces the following output

```
REASON:
To achieve the goal of clicking button ONE and then button TWO, we need to first
click button ONE to activate it, and then click button TWO to activate it.
ACTION:
CLICK 4
```

when the correct output would have been `CLICK 5`, given that the previous actions show that `CLICK 4` has already been executed.

**2. Skipping over intermediate actions.** We see that LLaMA-13B skips over actions when faced with multistep tasks. For example, on a book flight task with browser content

```
... <h2 id=5 val=Book Your One-Way Flight />
<input_text id=7 val=flight-from /> ...
```

the model generates an action like `CLICK 5` directly before filling in the empty flight-from input box.

**3. Not following the desired action formats.** We observe that LLaMA-13B often times fails to follow the specified action formats for CLICK and TYPE. This problem occurs for both Flat and SteP.

For example, given the following browser content

```
... <div id=2 val=wrap />\n<link id=4 val=justo. />...
```

the models predict `CLICK #justo` instead of `CLICK 4`.

Similarly, the model predicts `TYPE "Ignacio"` instead of `TYPE 5 "Ignacio"`.

## C   Airline CRM Simulator

We constructed a mock Airline CRM environment based on typical airline call-center work-flows, and modelled on public airline websites (e.g., jetblue.com, aa.com). The website is built as a standard single-page web application, using a light-weight web development stack. Figs 12, 13 shows sample screenshots from the CRM.

### C.1   Generating scenarios

This CRM allows us to test scenarios that are more complex than those in MiniWoB++, and scenarios that cannot practically be run on public websites (e.g., cancelling a flight). The scenarios currently supported are described below.

1. **Find one-way or return flight.** This is a single step task, that requires the source & destination airports and flight dates to be entered in the UI, and the search button to be clicked.
2. **Book flight** This is a four step task:
   (a) Find flight (scenario 1),
   (b) Select desired outward and return flights & click *Confirm*,
   (c) Enter passenger details (*Title, first name, last name, gender, date-of-birth*) & click *Save*,
   (d) Enter payment card details (*card number, expiry, CVC/CVV*) & click *Book flight*.
3. **Find a booking** This is a single step task - enter booking reference & click *Search*.
4. **Cancel a booking** This is a three step task:
   (a) Find booking (scenario 3),
   (b) Click *Cancel*,
   (c) Confirm cancellation by re-entering booking reference & click *Cancel*.
5. **Modify passenger details on an existing booking** This is a three step task:
   (a) Find booking (scenario 3),
   (b) Click *Modify*,
   (c) Change any required passenger details & click *Save*.
6. **Modify flights on an existing booking** This is
   (a) Find booking (scenario 3),
   (b) Click *Modify*,
   (c) Find flight (scenario 1),
   (d) Select desired outward and return flights & click *Save*,

### C.2   Helper APIs

In addition to supporting the above scenarios, the CRM also exposes a few helper APIs that make running and evaluating experiments easier. Two of these are of interest here:

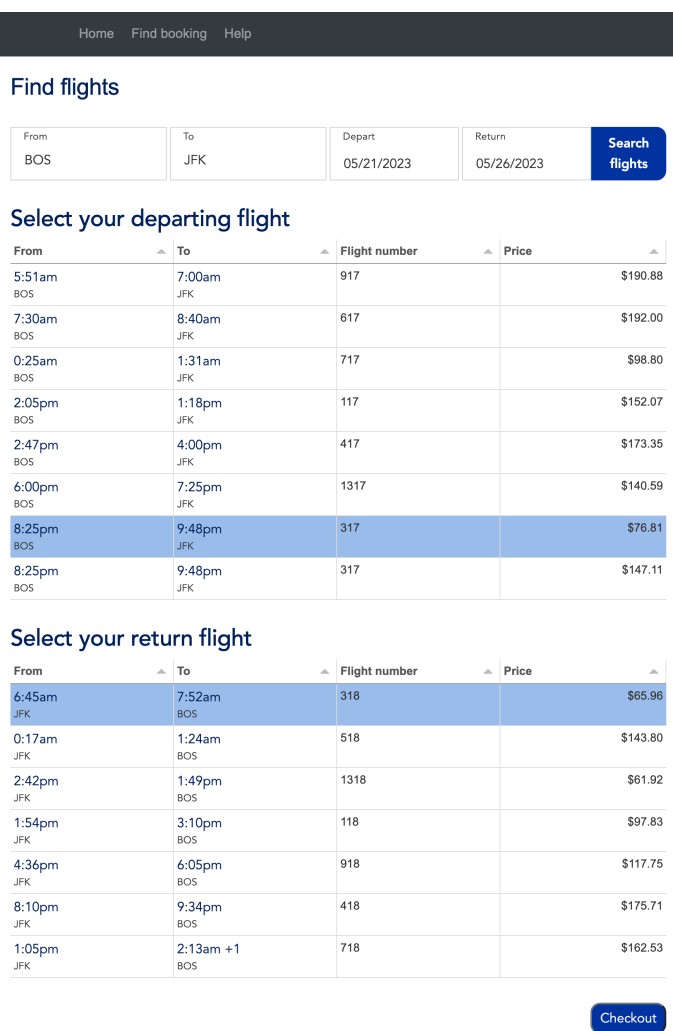

Figure 12: *search flight* screen of the mock airline CRM.

- `https://<base-url>/generate-random-scenario`. This API returns a scenario randomly selected from those listed above, along with all of the data required for completing that scenario on the UI. Shown below is an example of a scenario returned by this API. In addition to the task specific data, the scenario includes a unique id, and a unique URL on which the task can be executed.

```
1  {
2      "scenario": "TASK_FIND_FLIGHT",
3      "id": "ylmjd3iuqpdc3gdrvspq",
4      "url": "https://<base-url>/?scenario=ylmjd3iuqpdc3gdrvspq",
5      "details": {
6          "flight": {
7              "from": "JFK",
8              "to": "FLL",
9              "departure": "2023-07-07",
10             "return": "2023-09-13",
11             "outward-departure-time": "7:01pm",
12             "outward-arrival-time": "0:13pm",
13             "return-departure-time": "6:00am",
14             "return-arrival-time": "8:43am"
15         }
16     }
```

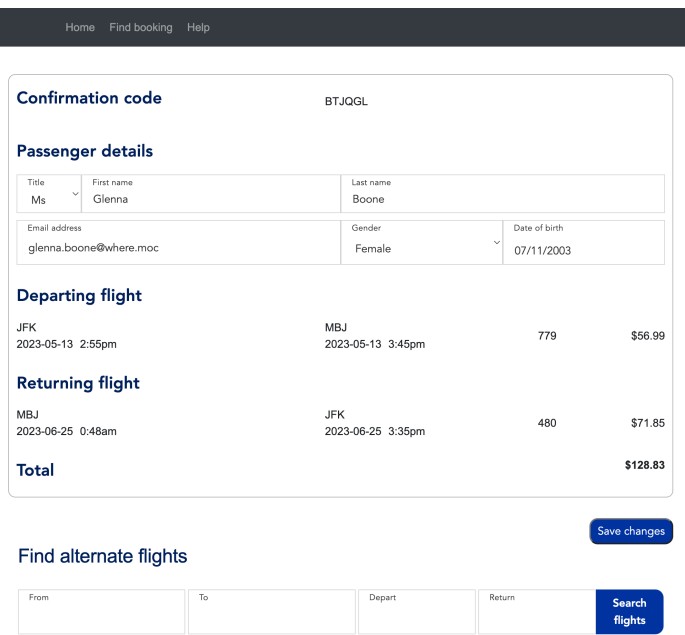

Figure 13: *find and modify booking* screen of the mock airline CRM.

```
17  }
```

- `https://<base-url>/evaluate?scenario=<id>` This API provides a means of automatically evaluating the *success rate* and *task progress* metrics for scenarios generated by the API above. Specifically, if the UI actions are performed on the unique URL returned by the *generate-random-scenario* API, calling the *evaluate* API afterwards with the scenario id will return the metrics. These metrics are calculated with reference to the gold standard actions required for the given scenario.

## D  Live Websites Evaluation

### D.1  Collecting Human Task Demos

We collected a dataset of human agents searching for flights across three websites: `https://www.jetblue.com/`, `https://www.aa.com/`, `https://www.united.com/en/us`. For each of the websites, a human agent was given a set of 10 task specifications as short conversations, e.g. "Book a flight departing from <>, arriving at <>, leaving on <> and returning on <>". The actions of the human agent, e.g. the click and types, were recorded for each episode. Along with the actions, the raw DOM of the website was also recorded.

### D.2  Parsing Browser Content

Given a data point of raw website DOM (Document Object Model) and action, we make use of playwright `https://playwright.dev` to parse the DOM and extract a list of web elements. This process involves traversing the DOM tree and extracting salient nodes, i.e. nodes with actionable elements like <input>, <button>, <link>. We also propagate up text attributes from child nodes to the salient parent nodes since the text label for an element may occur inside the subtree. Every

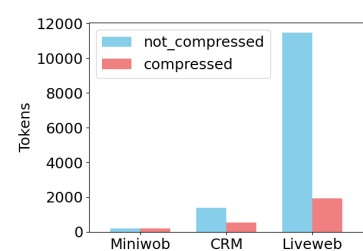

Figure 14: Token counts for browser content before and after compression on different environments.

web element in the list is represented by an `id` and
optionally a `value`. The list of all such elements
is concatenated to represent the browser content
in natural text form. This is input as the browser
observation in the LLM context. Natural language descriptions attached to different web
elements helps it generalize across different websites since LLMs have been pre-trained
on natural language data. This text form is included under Browser Content in the LLM
context. We also convert the demonstrated actions to `CLICK <id>` or `TYPE <id> "TEXT"`.

### D.3 Live Website Results

Fig. 15 shows evaluation of `SteP Few-shot` and `Flat Few-shot` across 10 runs each on 3
different live websites with task specification coming from short simulated conversations.
What makes this task challenging is that the browser content from these websites have a lot
of extraneous information that make it challenging to parse the correct fields. Fig. 14 shows
the extent of compression we perform to fit the browser content into the LLM's context space.
For each run, we evaluate by comparing model performance against a reference human
demonstration. In Fig. 15, `SteP Few-shot` is able to generalize to multiple websites even
though it has demonstration from only one (i.e. jetblue.com). In contrast, `Flat Few-shot`
fails to generalize from it's demonstration. Again `SteP Few-shot`, by decomposing the
problem into policies is able to solve the task.

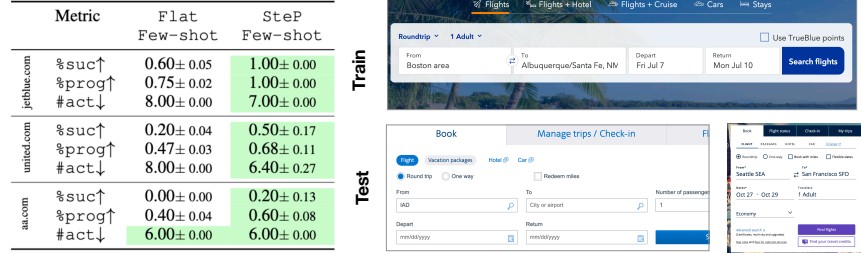

Figure 15: **(Left)** Evaluation on 3 live airline websites averaged over 10 runs per website. **(Right)**
Difference in train (jetblue) v/s test (united, aa) website UIs.

