# OpenReview forum: "SteP: Stacked LLM Policies for Web Actions"
_colmweb.org/COLM/2024/Conference — COLM_

### Official Review · Reviewer_BzFx · 2024-05-07

**Rating:** 8
**Confidence:** 2
**Ethics Flag:** 1

**Summary:**

They advance the research on the line of performing web tasks with LLMs by decomposing them into different policies (instead of using a single prompt). In particular, they develop an approach based on a Markov Decision Process where the state is compounded by a stack of policies, which represent the sequence of policy calls. Evaluation of their methodology across key benchmarks such as WebArena, MiniWoB++, and a CRM simulator reveals promising results, demonstrating efficiency in data consumption.

**Reasons To Accept:**

- The authors introduce an innovative method for addressing the challenge of executing web tasks with LLMs.
- They conduct a comprehensive comparison with other baseline methods across major benchmarks.
- Their approach demonstrates superior performance over the state-of-the-art in certain benchmarks, all while necessitating less data.
- The paper is well structured and presents information in a clear and accessible manner.
- Following acceptance, the code will be made publicly accessible.
- I believe it encompasses a satisfactory amount of work and quality to merit acceptance at COLM.

**Reasons To Reject:**

- The absence of a Conclusion section, which consolidates the conclusions drawn from preceding sections, is noted. However, aside from this observation, I do not identify any other issues with this paper.

---

> ### Author Rebuttal · Authors · 2024-05-30
>
> We thank the reviewer for their feedback!
>
> We will be sure to include a conclusion section containing the following key takeaways:
>
> * LLMs face significant challenges solving web tasks, including combinatorially large open-world tasks and variations across web interfaces. This often results in performance substantially lower than human performance.
> * Simply specifying a large prompt to handle all possible tasks and behaviors is complex, and results in behavior leaks between unrelated behaviors.
> * SteP successfully performs a diverse set of web tasks by decomposing them into distinct policies, each solving a specific subtask. At test time, it dynamically composes these policies to solve a new task.
> * Our key findings include,
>   * On WebArena, SteP outperforms (0.15 -> 0.36) prior works that use few-shot LLM (GPT-4) policies, while on MiniWob++, SteP is competitive with prior works while using significantly less data.
>   * SteP achieves this while using 2.3x lesser tokens per trajectory, resulting in lower overall costs.
> * SteP opens up possibilities for interesting future research, such as the potential to orchestrate models of varying complexity for different policies or exploring automatic policy discovery.

---

> > ### Comment · Reviewer_BzFx · 2024-06-05
> >
> > Thanks for your response. I'm sure that incorporating a conclusion section will enhance the final version of the paper.

---

### Official Review · Reviewer_TTFb · 2024-05-08

**Rating:** 6
**Confidence:** 4
**Ethics Flag:** 1

**Summary:**

This paper proposed a method called Step, which aims to solve the complex web navigation tasks. Instead of relying on one global policy/prompt to control how agent interact with the environment, Step contains a set of policies that are tailored to solve different sub-tasks, and the Step agent dynamically composes/invokes different policies to solve the task at hand. In particular, each policy can choose to execute an action in the environment, call other policies or call itself, which offers great flexibility in agent’s exploration. The Step method is evaluated on popular web agent benchmarks including WebArena, MiniWob++ and CRM and a few real-world websites. The results show that it outperforms previous methods and baselines with a flat policy.

**Questions To Authors:**

Please see the limitation section

**Reasons To Accept:**

The proposed method is very intuitive that the complex tasks can be decomposed into simpler ones, and each simple task can be handled by dedicated policies. The results on web agent benchmarks are also pretty promising.
Overall the Step agent uses less tokens in its prompt to complete the task compare to previous baselines, hence that it is more cost effective.

**Reasons To Reject:**

My biggest concern is about the construction of policies. As the authors admitted in the limitations, the Step method requires user to manually write detailed prompts for every policy. This implies that the method cannot be easily scaled to more websites/tasks. Moreover, from the examples shown in Appendix B.2 we can see that these policies/prompts are very tailored towards specific websites. For example, search order policy informs the agent to first go to My Account and then go to My orders. I’m not sure if the prompt can transfer well out of the box to different websites. For example, what if a website do not have the My Account button but the user actually needs to click their profile picture to get to the account details?
In section 5.1.2., the authors mention that the manully written policies coveres  at least 50 intent templates from WebArena. Have you checked if the gains on WebArena are mainly from these covered intents?

Missing references:
The idea of constructing specific policies/prompts at different states to control the web agent has been explored in Ma et al. 23. It would be good to compare your method with their work as well.

Ma, Kaixin, Hongming Zhang, Hongwei Wang, Xiaoman Pan and Dong Yu. “LASER: LLM Agent with State-Space Exploration for Web Navigation.” ArXiv abs/2309.08172 (2023): n. pag.

---

> ### Author Rebuttal · Authors · 2024-05-30
>
> We thank the reviewer for their feedback!
>
> **Manually crafted policies**
>
> While we agree that adding a new policy requires manually constructing a prompt, it is no more additional work than prompt design common in current LLM agents. E.g., total human-written instruction characters in SteP policy library is similar to existing Flat/ReAct web agents, Flat-4k (7.5k chars, 0.20 success) -> Flat-8k (17.3k chars, 0.23 success) -> SteP (20.2k chars, 0.36 success). We do not write policy prompts to solve individual tasks as that requires a combinatorially large number of policies. Rather, each policy solves a commonly occurring subtask shared across multiple tasks.
>
> While the current instructions may appear specific to websites in WebArena, it is easy to extend them to cover more websites. E.g. search_order() policy instructions (shared by 69 tasks) can be thought of as hints (not hard rules) to the LLM agent on how to locate the page containing order, e.g. My Account -> My orders, or click profile picture -> My Orders. Ultimately, the LLM agent chooses appropriate actions given webpage observation. We already do this for our LiveWebsite evaluation (Appx E) where the same choose_date() policy generalizes across different websites like JetBlue, American, United that have different UIs.
>
> We do observe that the ***majority of the gains (8.1% -> 47.9%) come on tasks when policies are invoked*** vs (19.2% -> 25.8%) when policies are not. Note that a library of only 14 policies covers 47.5% of the total 804 WebArena tasks, indicating our policies are not task-specific.
>
> We also note that our work is developed within the specific context of existing works designing prompts for LLM agents. Automatically discovering useful policies is an important, orthogonal problem not only to our work but to the general development in the field, outside our current scope.
>
> **Comparison to related work**
>
> Thank you for the reference, we will be sure to include it.
> * LASER define a finite state machine (FSM) where every state corresponds to an LLM agent performing a subtask. However, they do not hierarchically decompose a task into a set of subtasks. This would require tracking which agent called which other agent, not expressed in the FSM (but expressible in SteP as the policy stack)
> * SteP is evaluated on WebArena (5 domains) while LASER on WebShop (1 domain) making direct comparison difficult. We compare against prior works on WebArena, and ***achieve the top result on WebArena.***

---

> > ### Comment · Reviewer_TTFb · 2024-06-02
> >
> > Thanks for the reply.

---

### Official Review · Reviewer_piMY · 2024-05-11

**Rating:** 6
**Confidence:** 3
**Ethics Flag:** 1

**Summary:**

This paper proposes SteP, which dynamically composes policies from a policy library into a policy stack, to solve web tasks. The proposed model shows strong performance on WebArena, MiniWoB, and a CRM task.

**Questions To Authors:**

- What kind of prompts does the meta-policy need for each task, in order to manage the policy stack?
- Can the authors share more details about how the policy library was chosen on each task?

**Reasons To Accept:**

- The stack formulation for composing policies is intuitive and effective.
- The performance is strong on multiple tasks.
- The paper is clear and well written.

**Reasons To Reject:**

My main concerns are as follows:
- The task-specific nature of the proposed method, as it relies on a manually crafted choice of policy library, as well as demonstrations for the policies.
- A lack of novelty with respect to existing decompositional methods such as [1,2], as both methods explored the utilization of dynamic decomposition using a predefined skill library.

[1] Decomposed Prompting: A Modular Approach for Solving Complex Tasks. Khot et al., ICLR 2023
[2] ADaPT: As-Needed Decomposition and Planning with Language Models. Prasad et al., NAACL 2024

---

> ### Author Rebuttal · Authors · 2024-05-30
>
> We thank the reviewer for their feedback!
>
> **Task-specific, manual policy library**
>
> While we agree that adding a new policy requires manually constructing a prompt, it is no more additional work than prompt design common in current LLM agents. E.g., total human-written instruction characters in SteP policy library is similar to existing Flat/ReAct web agents, ***Flat-4k (7.5k chars, 0.20 success) -> Flat-8k (17.3k chars, 0.23 success) -> SteP (20.2k chars, 0.36 success)***
>
> We don't write policy prompts to solve individual tasks as that requires a combinatorially large number of policies. Rather, each policy solves common subtasks shared across multiple tasks. E.g., ***SteP uses only 14 policies on 804 WebArena tasks***
>
> SteP's key contribution is its dynamic composition of policies, allowing policies to decide which other policies to call. This expands task coverage beyond the instructions in policy prompt and enables the LLM to contextually load instructions for specific subtasks.
>
> Our work is developed within the context of existing works designing prompts for LLM agents. Discovering useful policies or skills autonomously (a long standing challenge in hierarchical RL [3]) is an important but orthogonal problem to the general development of the field, outside our current scope.
>
> **Comparison to [1,2]**
>
> We cover [1,2] in related work. Both don't address web automation, other important differences are:
>
> DecomP [1] vs SteP:
> * DecomP subroutines do not call each other (except for recursions), restricting it to a static hierarchy. SteP allows any policy to call any other, creating a dynamic hierarchy.
> * DecomP generates a static program upfront, whereas SteP does this dynamically, changing the policy call sequences based on observations.
>
> ADaPT [2] vs SteP:
> * ADaPT predicts a plan and greedily decomposes it until one of the subtasks can be solved, without backtracking.
> * SteP uses policies instead of plans – allowing it to react to failures quicker by changing the decomposition.
>
> **Prompts to manage policy stack?**
>
> The stack is managed outside of the LLM. We simply provide the LLM a list of available policies to choose from.
>
> **How was policy library chosen?**
>
> Each policy is designed by identifying commonly occurring subtasks from related task clusters, e.g. search_order() policy is shared by 69 tasks (20 clusters) like “Change delivery address..”, “Draft refund message..”
>
> ***
> [3] Diversity is All You Need: Learning Skills without a Reward Function

---

> > ### Comment · Reviewer_piMY · 2024-06-05
> > **Response to authors**
> >
> > I appreciate the authors' response. I understand the distinction from DecomP and ADaPT better now. As I understand it, it offers a much more flexible control of policy composition than those methods.
> >
> > That said, in order to be complete, I think paper needs to include the full policy library for all tasks, since these are all handcrafted for each task.
> >
> > Overall, I think the method is interesting and I have increased my score accordingly.

---

> ### Author Response · Authors · 2024-06-05
>
> Thanks so much for your feedback. We'll be sure to extend the 5 policy prompts in Appendix B to include the remaining prompts from the full policy library.

---

### Decision · Program_Chairs · 2024-07-10

**Decision:**

Accept

**Comment:**

The paper presents SteP, a novel approach for web task automation that dynamically composes policies from a policy library into a policy stack. The reviewers agree that the proposed method demonstrates significant improvements over existing methods on benchmarks such as WebArena, MiniWoB++, and a CRM task. The intuitive and effective stack formulation, combined with strong performance results, highlights the method's potential. However, concerns were raised about the task-specific nature of the policy library and the manual effort required to construct it. The authors' rebuttal clarified the scalability and flexibility of the policy design and distinguished their approach from similar works. Overall, the paper is well-written and contributes valuable insights to the field, meriting acceptance.